# TOWARDS GENERAL ROBUSTNESS TO BAD TRAINING DATA

## ABSTRACT

In this paper, we focus on the problem of identifying bad training data when the underlying cause is unknown in advance. Our key insight is that regardless of how bad data are generated, they tend to contribute little to training a model with good prediction performance or more generally, to some utility function of the data analyst. We formulate the problem of good/bad data selection as utility optimization. We propose a theoretical framework for evaluating the worst-case performance of data selection heuristics. Remarkably, our results show that the popular heuristic based on the Shapley value may choose the worst data subset in certain practical scenarios, which sheds lights on its large performance variation observed empirically in the past work. We then develop an algorithmic framework, DATASIFTER, to detect a variety of and even unknown data issues—a step towards general robustness to bad training data. DATASIFTER is guided by the theoretically optimal solution to data selection and is made practical by the data utility learning technique. Our evaluation shows that DATASIFTER achieves and most often significantly improves the state-of-the-art performance over a wide range of tasks, including backdoor, poison, noisy/mislabel data detection, data summarization, and data debiasing.

## 1 INTRODUCTION

The quality of training data is a fundamental ingredient towards useful and reliable ML-based applications. Unfortunately, there are unaccountably many possible data issue types. For example, errors and bias occur frequently in data generation and collection processes. Bad data could also be caused by adversarial attacks (e.g., data poisoning and backdoor attacks), as training data are often collected from anonymous and unverified sources. On the other hand, most of the existing data selection strategies only applicable to specific data issues. Due to the diversity of data issues, there is an urgent need to achieve *general robustness* to bad data of various types and even unknown types.

Purging bad data is a long-standing problem, intensively studied by both the database and the ML community. Most of the existing approaches, however, can only achieve *specialized robustness* to bad data of certain types. In the database community, the state-of-the-art data cleaning approaches such as ActiveClean (Krishnan et al., 2016), BoostClean (Krishnan et al., 2017), AlphaClean (Krishnan and Wu, 2019), and CPClean (Karlaš et al., 2020) are only applicable to certain classes of ML models and data issues. In particular, they cannot detect adversarial attacks on training data. The endeavors of the ML community have covered a wider range of data quality issues such as adversarial attacks (Wang et al., 2019; Chen et al., 2019), data debiasing (Zemel et al., 2013; Madras et al., 2018), and mislabel detection (Zhao et al., 2019). However, each aforementioned solution only focuses on addressing a specific data issue effectively. The effectiveness of these approaches is based on the premise that the data quality issues are known *a priori*, which usually does not hold in reality. Few recent works have the potential to achieve general robustness to unknown data issues. One line of such works is based on differentially private training (Du et al., 2019; Hong et al., 2020). These approaches do not perform data selection; instead, they attempt to restrict the impact of each training data point on the learning outcome in an undifferentiated manner. As a side effect, these approaches hinder learning from good data, thereby leading to poor learning performance in practice (Tramèr and Boneh, 2020). Another line of works that has potential to achieve general robustness is based on data valuation. These works first adopt a data importance metric, e.g., influence function (Koh and Liang, 2017; Koh et al., 2019; Feldman and Zhang, 2020) or Shapley value (Ghorbani and Zou, 2019; Jia et al.,

2019a), to quantify each training data point's contribution to the training process. Then, which data to retain or remove is decided based on the *ranking of data value*. However, past empirical studies (Jia et al., 2019b) and our experiments (Section 6) show that their performance varies considerably across different data and learning algorithms. Overall, existing approaches to bad data filtering either cannot identify unknown data issues by design or suffer from poor detection efficacy.

In this work, we take a step towards general robustness to bad training data. Our work is underpinned by a key insight about what defines "bad training data". Despite the diverse types of data issues, the crucial commonality is that all bad data contribute little to achieving good model performance. If a data point contributes positively to learning, it would be beneficial to just keep it as part of the training set, thus not considered a "bad" point. Hence, a promising way to select data without knowing data issue types in advance is to search for data subsets that results in the highest trained model performance. More generally, the data analyst may have a utility function beyond model performance, so we formulate data selection as a utility optimization problem (Section 3).

Secondly, with the utility optimization objective in mind, we present a novel theoretical framework (Section 4) for rigorously analyzing the worst-case performance of data selection approaches. The line of existing works closest to achieving general robustness is the aforementioned data value ranking approaches. However, we show that these approaches have unsatisfying worst-case performance guarantees due to failure to capture the interactions amongst selected data points (also empirically shown in Appendix F.5.2). In particular, the popular Shapley value-based approach could select the worst data in some common scenarios.

We then design a general algorithmic framework guided by the theoretically optimal (but computationally infeasible) solution to utility optimization (Section 5). A significant technical challenge for finding the subset that optimize the model utility is that, in order to evaluate the impact of different subsets of data on model performance, we need to retrain model on every possible subset. To solve the computational challenge, we introduce DATASIFTER, which directly learns a parametric function to predict the performance of a model trained on a given subset and then performs data selection via optimizing the function. Compared with prior data selection algorithms, DATASIFTER has the following advantages: (1) being able to handle various data issues (*general robustness*), (2) applies to any target ML model architectures (*model-agnostic*), and (3) instantiated by the goal of downstream ML tasks (*task-driven*). Finally, we conduct a thorough empirical study on a range of data issues (Section 6), including backdoor and poisoning attack detection, noisy label/feature detection, data summarization, and data debiasing. Our experiments demonstrate that DATASIFTER achieves and most often significantly improves the state-of-the-art performance of data valuation-based approaches.

## 2 RELATED WORK

*Data valuation-based* approaches can potentially achieve general robustness against diverse data issues by quantifying data importance or "data value", and then picking data points with high value for model training process. One simple idea to quantify data importance is to use the leave-one-out error. Koh and Liang (2017) provides an efficient algorithm to approximate leave-one-out error for each training point. Recent works leverage credit allocation schemes originated from cooperative game theory to quantify data importance. Particularly, Shapley value has been widely used (Ghorbani and Zou, 2019; Jia et al., 2019a;c;b; Wang et al., 2020), as it uniquely satisfies a set of desirable axiomatic properties. More recently, Yan and Procaccia (2020) suggests that the Least core is also a viable alternative to Shapley value for measuring data importance. However, computing the exact Shapley and Least core values are generally NP-hard. Several approximation heuristics, such as TMC-Shapley (Ghorbani and Zou, 2019), G-Shapley (Ghorbani and Zou, 2019), KNN-Shapley (Jia et al., 2019c), have been proposed for the Shapley value. Despite their computational advantage, they are biased in nature. On the other hand, unbiased estimators such as Permutation Sampling (Maleki, 2015) and Group Testing (Jia et al., 2019a) still require retraining models many times for any decent approximation accuracy. TracIn (Pruthi et al., 2020) estimates the importance by tracing the test loss change caused by a training example during the training process. The representer point method (Yeh et al., 2018) captures the importance of training point by decomposing the pre-activation prediction of a neural network into a linear combination of activations of training points.

The typical paradigm of data valuation-based approaches for selecting high-quality data (or filtering bad data) is straightforward: (1) each data point's value (e.g., Shapley value) is computed. (2) data

points are sorted by value and data points with the highest value are selected. However, this paradigm fails capture the interactions amongst selected data points (Section 4 and Appendix F.5.2), i.e., the existence of one data point often affect the importance of another.

*Differentially private training* (Du et al., 2019; Hong et al., 2020) have the potential to achieve general robustness to unknown data issues. However, fundamentally, these approaches diminish the influence of each training data on the learned model in an undifferentiated manner. Hence, these approaches hinder learning from good data, which lead to poor learning performance (Tramèr and Boneh, 2020).

*Data cleaning approaches* from database community are not able to achieve general robustness. The state-of-the-art data cleaning approaches leverage the information about downstream ML tasks to guide the cleaning process; examples include ActiveClean (Krishnan et al., 2016), BoostClean (Krishnan et al., 2017), AlphaClean (Krishnan and Wu, 2019), and CPClean (Karlaš et al., 2020). However, the state-of-the-art data cleaning methods are only applicable to certain architectures of ML models (e.g., convex model, nearest neighbors), data format (e.g., tabular data), and data issues (e.g., missing values, outliers). In particular, data cleaning approaches cannot be straightforwardly extended to adversarial attacks such as data poisoning attacks and backdoor attacks.

## 3 FORMALISM

We propose to formulate the problem of achieving general robustness to bad data as finding a subset of data points with the highest utility. We use the data utility function to characterize the mapping from a set of data points to its utility.

More formally, let $\mathcal{D} = \{(x_i, y_i)\}_{i=1}^n$ denote the training set with data points of different quality. A learning algorithm $\mathcal{A}$ is a function that takes a dataset $S \subseteq \mathcal{D}$ and outputs a classifier $\hat{f}$. A metric function $u$ takes $\hat{f}$ as input and outputs its model utility. In the machine learning context, we often use test accuracy as the metric, $u(\hat{f}, \mathcal{V}) = \frac{1}{|\mathcal{V}|} \sum_{(x,y) \in \mathcal{V}} \mathbb{1}[\hat{f}(x) = y]$ for a test set $\mathcal{V}$. However, test set $\mathcal{V}$ is usually not available during the training time. In practice, $u(\hat{f}, \mathcal{V})$ is typically approximated by *validation accuracy* $u(\hat{f}, V)$ where $V$ is a validation set separated from the training set.

With a potentially randomized learning algorithm $\mathcal{A}$ and a corresponding metric function $u$, we define the data utility function as $U_{\mathcal{A},u}(S) = \mathbb{E}_{\mathcal{A}} [u(\mathcal{A}(S), \mathcal{V})]$. When the context is clear, we omit the subscript and simply write $U(S)$. The concept of data utility functions was originally discussed in Wang et al. (2021a), where it is used for active learning tasks. The critical difference between our formulation and Wang et al. (2021a) is that our formulation incorporates the label information as the input to a data utility function. This is important as label information is required for identifying both mislabeled and many types of adversarial attacks on training data.

With the modified notion of the data utility function, we abstract the objective of selecting high-quality data as a utility optimization problem:

$$\max_{S \subseteq \mathcal{D}: |S| = k} U(S) \tag{1}$$

where $0 < k < n$ indicates the selection budget, which can be predetermined (e.g., based on the prior knowledge about potential data defects or computational requirements). The fundamental intuition behind this formulation is that despite the diversity of data issue types, all "bad data" have the commonality that they all contribute little or negatively to model performance. Hence, optimizing data utility function is a principled way to achieve general robustness. Besides, the interactions between data points will significantly affect model performance. A "good data" can only be defined relative to the rest of the data points in the dataset. Therefore, high-quality data selection needs to be done in a batch-style instead of one-by-one. Overall, a promising way to deal with unknown data issues is to search for a *set* of data points that results in the highest trained model performance on a clean validation set.

## 4 WORST-CASE ANALYSIS FOR VALUATION-BASED APPROACHES

In this section, we introduce a theoretical framework to analyze the worst-case performance of data selection algorithms for the utility optimization objective in (1). We show that data valuation-based approaches, such as *leave-one-out* (LOO), Shapley value, and Least core, achieve unsatisfying

worst-case guarantee. Note that here we assume exact data value can be computed for every data point. In practice, however, data value notions such as Shapley value require model retraining on every possible data subset. Typically, only approximations can be obtained, which may further impair data selection results. We will defer all proofs to Appendix.

We start by formalizing data selection algorithms into a general paradigm. We call an algorithm $\mathcal{M}$ such that $\mathcal{M}((\mathcal{D}, U), k)$ returns $S \subseteq \mathcal{D}$ of size $k$ a *heuristic* to a (size-$k$) data selection problem on dataset $\mathcal{D}$. The typical pattern of data valuation-based heuristics is that they first rank the data points according to their corresponding data importance metric, and then select the data points with the highest importance scores. We call the heuristics matching this selection pattern as *linear heuristics*.

**Definition 1** (Linear heuristic). *We say $\mathcal{M}$ is a linear heuristic for data selection problem if for every instance $\mathcal{I} = (\mathcal{D}, U)$ where $\mathcal{D} = \{z_1, \dots, z_n\}$, $\mathcal{M}$ works as follows:*

1. *Assign a score $v = (v_1, \dots, v_n)$ for every data point $z_i \in \mathcal{D}$. Sort $\mathcal{D}$ according to $v$ in non-ascending order and get $(z_{(1)}, \dots, z_{(n)})$. Certain rules are applied to break tie.*

2. *For a query of selecting $k$ high-quality data points, return the $k$ data points in $\mathcal{D}$ with the highest scores $(z_{(1)}, z_{(2)}, \dots, z_{(k)})$.*

For example, for Shapely-based data selection approach, each $v_i = \sum_{S \subseteq D \setminus \{z_i\}} \frac{1}{n\binom{n-1}{|S|}} \big[ U(S \cup \{z_i\}) - U(S) \big]$ and $\mathcal{M}((\mathcal{D}, U), k)$ returns $k$ data points with the highest Shapley values.

Our theoretical framework for studying the worst-case performance of data selection heuristics is inspired by the domination analysis framework initially proposed in Glover and Punnen (1997). Our worst-case performance metric is *domination ratio*, which measures how many subsets achieve lower utility than the selected set in the worst-case scenario.

**Definition 2** (Domination ratio). *The domination ratio of a heuristic $\mathcal{M}$ for the data selection problem is the maximum ratio $0 < d(n, k) \leq 1$ s.t., for every problem instance $\mathcal{I} = (\mathcal{D}, U)$ on a dataset $\mathcal{D}$ of size $n$ and utility function $U$, $\mathcal{M}(\mathcal{I}, k)$ produces a size-$k$ subset $S \subseteq \mathcal{D}$ which has utility $U(S)$ no worse than at least $d(n, k)$ percentage of all size-$k$ subsets.*

For example, $d(n, n) = 1$ for any heuristic since there is only one choice of size-$n$ set. $d(n, n/2)$ will be calculated by the *least* number of size-$n/2$ subsets $S \subset \mathcal{D}$ where $\mathcal{M}((\mathcal{D}, U), k)$ has better utility than across all datasets $\mathcal{D}$ and utility function $U$, divided by the total number of subsets $\binom{n}{n/2}$. The domination ratio is well defined for every data selection heuristic. A heuristic with a higher domination ratio may be a better choice than a heuristic with a smaller domination ratio due to the better worst-case guarantee. The best heuristic for data selection has domination ratio $d(n, k) = 1$ for every $k \leq n$, which means that it will always pick the size-$k$ data subset with the highest utility (the exact solution for Objective (1)) for every possible data utility function. The worst possible domination ratio is $d(n, k) = 1/\binom{n}{k}$, as the returned subset has utility at least no worse than itself.

The following result shows that no linear heuristic is among the best heuristic whenever $n \geq 3$.

**Theorem 1.** *For $n \geq 3$, there exists no linear heuristic $\mathcal{M}$ s.t. $d(n, k) = 1$ for every $k \in \{1, \dots, n\}$.*

Furthermore, we can tighten the upper bound of the domination ratio for data valuation-based heuristics by noticing another common property: two data points will receive the same importance score if they contribute equally to all possible subsets of the training data. This property is often referred to as *symmetry axiom* in the literature (Jia et al., 2019a; Yan and Procaccia, 2020).

**Definition 3** (Symmetry axiom). *A linear heuristic $\mathcal{M}$ satisfies symmetry axiom if its scoring vector $v$ in the Step 1 of Def. 1 satisfies: $[(\forall S \in \mathcal{D} \setminus \{z_i, z_j\}) U(S \cup \{z_i\}) = U(S \cup \{z_j\})] \implies v_i = v_j$.*

The symmetry axiom may be desired for application scenarios requiring fairness, e.g., data importance scores are used to assign monetary rewards for data sharing. However, for data selection, symmetry axiom may be undesirable because simply gathering high-value data points may lead to a set of redundant points. Based on this intuition, the following theorem further upper bounds the domination ratio for non-trivial linear heuristics that with symmetry property.

**Theorem 2.** *If a linear heuristic $\mathcal{M}$ assigns different scores to different data points and satisfies symmetry axiom, then the domination ratio $d(n, k)$ of $\mathcal{M}$ is upper bounded by $\lfloor n/k \rfloor \binom{\lceil \frac{n}{\lfloor n/k \rfloor} \rceil}{k} / \binom{n}{k}$. In particular, when $c = n/k$ for some constant integer $c$, $d(n, k) \leq (k/n)^{k-1} = (1/c)^{O(n)}$.*

The above theorem suggests that whenever $k \leq n/2$, the domination ratio will be exponentially small for linear heuristics that satisfy symmetry axiom. The key insight is that symmetry axiom makes the heuristics tend to select similar data points as similar data points will have similar value, while the data utility does not simply add up. We experimentally demonstrate this in Appendix F.5.2.

Notably, we show that the Shapley value-based heuristic has no performance guarantee in the worst case scenario, even if we restrict data utility functions to be submodular, a common assumption of data utilities (Wang et al., 2021a;b; Han et al., 2020). The intuition is that Shapley value of training data weights higher for its marginal contributions on small datasets. Thus, data points that make a larger contribution on tiny datasets may be assigned with higher Shapley value, even if they make little or negative contributions in every dataset of desired selection size $k$.

**Theorem 3.** *For any $n \geq 4$ and $k \in \{1, \ldots, n\}$, the domination ratio of Shapley value-based heuristics is $d(n, k) = 1/\binom{n}{k}$, even if the data utility function $U$ is submodular.*

## 5 DATASIFTER

One straightforward way to optimize Objective (1) is to exhaustively evaluate $U(S)$ for all possible size-$k$ subsets $S \subseteq \mathcal{D}$ and choose the one that achieves the highest utility. Of course, this theoretically optimal but unrealistic algorithm requires prohibitively large computational resources as the required number of utility evaluations is $\binom{n}{k}$, and worse yet, each evaluation of data utility function requires retraining the model. Our approach to resolving the computation issues is inspired by a recent line of work, which shows that data utility functions for many common ML algorithms exhibit "approximate" submodularity. This property allows data utility functions to be learned (Balcan and Harvey, 2011; Wang et al., 2021c) and optimized (Minoux, 1978) efficiently. Hence, the idea of our approach is to first learn a parametric model to approximate data utility functions. With such a model, one can estimate the learning performance for any dataset by feeding the dataset at the input and query the function's output. Then, one can select the data subset by optimizing this model. If the model can fully recover the utility function $U$, under the same unbounded computation assumption made for the analysis of linear heuristics, the domination ratio is just 1 as our method can always numerate all possible subsets and find the optimal one.

Specifically, the proposed data selection framework, termed DATASIFTER, proceeds in two phases: *learning* and *selection* phase.

**Learning Phase.** Figure 1 depicts the learning phase of the DATASIFTER, which consists of a utility sampling step and a utility model training step. Formally, suppose that we have training set $\mathcal{D} = \{(x_i, y_i)\}_{i=1}^n$ and a small validation set $V$ representative for potential test data. To learn the data utility model, we train the classifier $\hat{f}$ for multiple times with different subsets $S \subseteq \mathcal{D}$. The set $\{(S, u(\hat{f}, V))\}$ could serve as a training set for learning $U$. When the learning algorithm is stochastic, for the sake of efficiency we just train $\hat{f}$ once and calculate $u(\hat{f}, V)$ to approximate the utility $U(S)$. If computational budget permits, one can choose to retrain $\hat{f}$ multiple times and compute the average utility. We adopt a canonical model architecture for set function learning–DeepSets (Zaheer et al., 2017)–as our model for $U$. A DeepSets model is a set function $f(S) = \rho\left(\sum_{z \in S} \phi(z)\right)$ where both $\rho$ and $\phi$ are neural networks, and in our context $z = (x, y)$ the concatenation of data feature and label. Its permutation-invariant property and universal expressive power making it suitable as the parametric model for learning data utility function. With the utility samples $\{(S, u(\hat{f}, V))\}$ as the training data, we can learn the data utility function through training the DeepSets model $\hat{U}$ with standard stochastic gradient descent. In this work, the sampling distribution of $S$ is simple: we first uniformly pick a set size, and then uniformly sample a subset of the given size without replacement. We stress that more fine-grained subset sampling methods could potentially improve the resulting data selection performance (see further discussion in Section 6.2.2 VI). We left exploring the relationship between subset sampling distribution and data selection performance as future work.

**Selection Phase.** Given a trained utility model $\hat{U}$, we could optimize it as a surrogate for objective (1) efficiently and approximately using greedy algorithms, as most of the data utility functions are empirically shown to be approximately submodular. We follow Wang et al. (2021a) and choose *stochastic greedy algorithm* (SG) from Mirzasoleiman et al. (2015) to find the subset that optimizes utility model (i.e., the trained DeepSets model). The SG is a simple algorithm that, for each iteration, randomly selects a subset of data $\{z_i\}_{i=1}^r$ and then finds the best data point within that subset. In the context of optimizing utility model, the "best data point" within each randomly selected subset refers

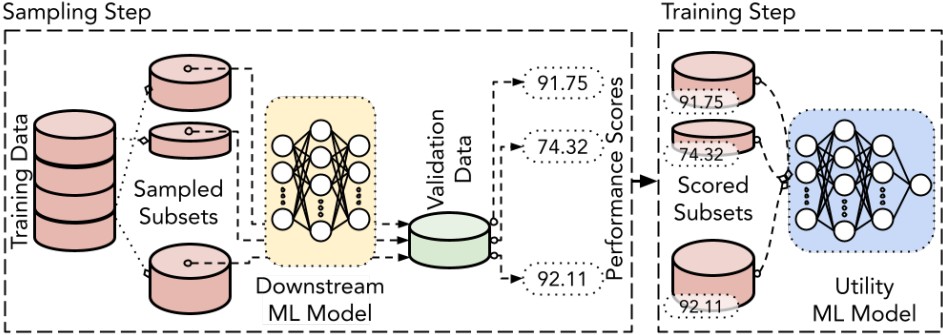

Figure 1: Overview of the learning phase, which consists of utility sampling step and utility model training. We randomly sample subsets from the training data during the sampling step, retrain the model on the subset and obtain the utility score for each set by evaluating the ML model over a clean validation set. Then, we train the utility model to predict the utility of a given dataset.

to the data point $z$ with the highest marginal contribution $\hat{U}(S \cup \{z\}) - \hat{U}(S)$, where $S$ is the set of data points selected in previous iterations. SG runs in linear time while provides decent optimization guarantee for submodular functions. Although its approximation guarantee has been proved only for monotone submodular objective functions, our experiments show that it achieves high empirical performance on data utility models that are approximately submodular. A more detailed discussion and pseudo-code for SG can be found in Appendix D. We also note that the rigorous characterization of the (approximate) submodularity for data utility functions is surprisingly difficult and it is still an open question (Wang et al., 2021a;b).

In our experiment (Section 6), the number of utility samples is 4000 for all the tasks we evaluated on (an empirically tuned number for efficiency-performance tradeoff). Model retraining could be computationally expensive for large training datasets and models. However, retraining on different subsets is common and often unavoidable for data selections algorithms aiming at general robustness, e.g., permuta-

| Method Type | General Robustness | Model-Agnostic | Task-Driven | Est. Utility |
|---|---|---|---|---|
| **Data Cleaning** | × | ○ | ○ | × |
| **Perm-SV (Maleki, 2015)** | ✓ | ✓ | ✓ | × |
| **TMC-SV (Ghorbani and Zou, 2019)** | ✓ | ✓ | ✓ | × |
| **G-SV (Ghorbani and Zou, 2019)** | ✓ | × | ✓ | × |
| **KNN-SV (Jia et al., 2019c)** | × | ✓ | × | × |
| **Least Core (Yan and Procaccia, 2020)** | ✓ | ✓ | ✓ | × |
| **Leave-one-out (Koh and Liang, 2017)** | ✓ | ✓ | ✓ | × |
| **Infl. Func. (Koh and Liang, 2017)** | × | × | ✓ | × |
| **TracIn (Pruthi et al., 2020)** | × | × | ✓ | × |
| **DATASIFTER** | ✓ | ✓ | ✓ | ✓ |

Table 1: Summary of the differences between prior works and DATASIFTER. ○ means only some of the techniques in the type satisfy the property. 'SV' means Shapley value.

tion sampling for Shapley value requires $O(n \log n)$ times model retraining. For DATASIFTER, two techniques could be used to improve scalability. The first technique is based on the observation that 4000 times of model retraining is still relatively efficient on small datasets. For example, training 4000 small CNN models on the *subsets of* 2000 CIFAR-10 images takes only about 15 hours with NVIDIA Tesla K80 GPU. As long as different data subsets have observable differences in utilities (e.g., 10% vs 30% test accuracy), the learned utility model can still be used to differentiate between good and bad data points. Since the learned utility model can provide utility estimations for sets of unseen data points by design, when the utility learning is on smaller subsets, greedy optimization can still be performed on a larger dataset by selecting good data in a batch-mode style. This largely improves the scalability of DATASIFTER (Appendix F.5.9). The second technique is to use a smaller proxy model for utility sampling, which is proposed in Wang et al. (2021a), and the idea of using proxy model for data selection was also explored in Lewis and Catlett (1994); Coleman et al. (2019). This technique is based on the observation that the utilities of data for different learning algorithms are usually *positively correlated*. In our experiment, we find that sampling utilities on small subsets can already achieve good efficiency and performance (Appendix F.5.9).

Compared with prior data selection algorithms mentioned in Section 2, DATASIFTER has the following advantages: (1) achieves general robustness by design, (2) is model-agnostic as data utility function can be defined for every learning algorithm, and (3) is task-driven as the optimization objective (the

utility function) is defined in terms of the downstream ML task (the learning algorithm and metric function). In addition, with the learned data utility model, DATASIFTER can provide an estimate of the utility for the selected dataset, which will be useful for data analysts to decide the number of data points to select. The major differences between DATASIFTER and previous data selection algorithms are summarized in Table 1.

## 6   EVALUATION

To test the general robustness of DATASIFTER, we evaluate this algorithmic paradigm on a variety of tasks with different data issues, as listed in Table 2. We consider various benchmark models and datasets used in past literature for each task. Since we observe similar results on different datasets, we only describe the result on *one* representative dataset for each task here and leave the other dataset in the Appendix. We also evaluate the scalability of DATASIFTER on larger datasets in Appendix F.5.9.

| Task | Datasets | |
|------|----------|----------|
| | Main Text | Appendix |
| **I.** Backdoor Detection | CIFAR-10 | MNIST |
| **II.** Poisoned Data Detection | CIFAR-10 | Dog vs. Cat |
| **III.** Noisy Feature Detection | CIFAR-10 | MNIST |
| **IV.** Mislabeling Detection | SPAM | CIFAR-10 |
| **V.** Data Summarization | PubFig83 | COVID-CT |
| **VI.** Data Debiasing | Adult | COMPAS |

Table 2: Summary of tasks and datasets.

### 6.1   IMPORTANT SETTINGS AND BASELINES

For fair comparison between DATASIFTER and baselines, we fix the number of utility sampling as 4000 for DATASIFTER as well as baseline algorithms that require utility sampling. For DATASIFTER, subsets are sampled by first picking a set size uniformly, and then uniformly sample a subset of the given size. The validation data in utility sampling are 500 clean data points sampled from the test data of the corresponding datasets. We repeat model training ten times for each selected set of data points to obtain the error bars. The metric function we use for Data Debiasing experiment is weighted accuracy, and for all other experiments the metric function is validation accuracy. Both DATASIFTER and baselines use the same metric function.

We focus on comparing data valuation-based approaches as they are closest to achieving general robustness. Differentially private training methods are omitted from comparison as they significantly impair model performance. Data cleaning methods are also omitted as their applicability is limited to specific data types, error types, and model architectures. We consider the following eight state-of-art data valuation-based approaches: (1) *Shapley Permutation Sampling* (**Perm-SV**) (Maleki, 2015), a Monte Carlo-based algorithm for Shapley value estimation. (2) *TMC-Shapley* (**TMC-SV**) (Ghorbani and Zou, 2019), a refined version of the Perm-SV, where the computation focuses on the subsets whose utility changes significantly when an extra point is added. (3) *G-Shapley* (**G-SV**) (Ghorbani and Zou, 2019), which approximates the Shapley value by anticipating the utility change caused by an extra point with its gradient. (4) *KNN-Shapley* (**KNN-SV**) (Jia et al., 2019c), which approximates the Shapley value by using K-Nearest-Neighbor as a proxy model. (5) *Least Core* (**LC**) (Yan and Procaccia, 2020), another data value notion in cooperative game theory with Monte Carlo-based approximation. (6) *Leave-one-out* (**LOO**) (Ghorbani and Zou, 2019) evaluates the change of model performance when a data point is removed. (7) *Influence Function* (**INF**) (Koh and Liang, 2017; Koh et al., 2019), which approximates the LOO error with first-order extrapolation. (8) **TracIn-Clean** (Pruthi et al., 2020), which traces the loss on clean validation data change during the training process whenever the training point of interest is sampled in batch SGD. (9) **TracIn-Self** (Pruthi et al., 2020) use the similar technique as TracIn-Clean but traces the self-influence, i.e., the reduction of a training point on its own loss. This approach is used for detecting bad data. (10) **Random** is a setting where we randomly select a subset from the target dataset.

### 6.2   RESULTS

#### 6.2.1   FILTERING OUT HARMFUL DATA

Training data could be contaminated by various harmful examples, e.g., backdoor triggers, poison information, noisy/mislabeled samples. Our goal here is to identify data points that are *most likely* to be harmful. These points can either be discarded or presented with high priorities to human experts for manual cleaning. To evaluate the performance of different data selection methods, we examine the training instances filtered by each method and plot the change of the fraction of detected corrupted

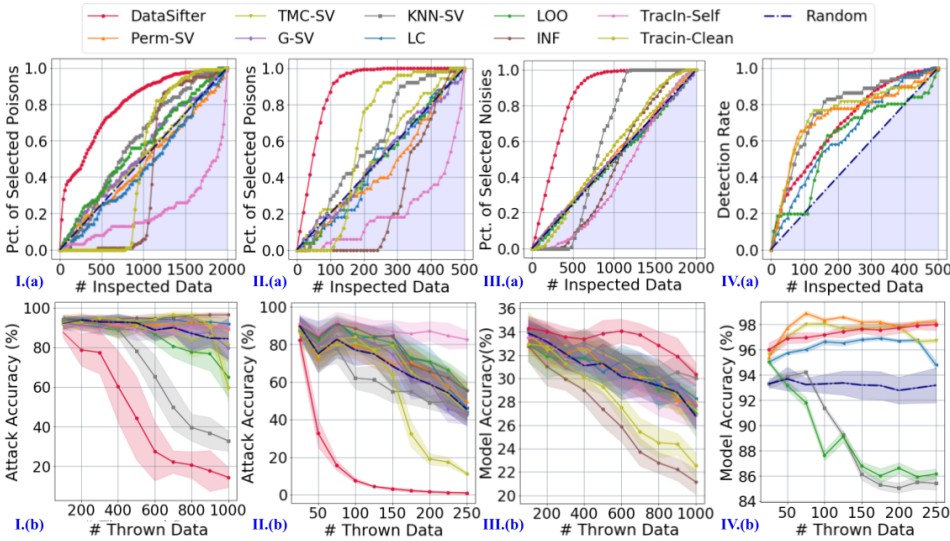

Figure 2: The experimental results and comparisons of the DATASIFTER under the case of filtering out harmful data (application I-IV). The light blue region in each (a) graph represents the area that a method is no better than a random selection. For I.(b) and II.(b), we depict the Attack Success Rate (ASR), where a lower ASR indicates a more effective detection. For III.(b) and IV.(b), we show the model test accuracy, where a higher accuracy means a better selection.

data with the fraction of the checked training data. Additionally, for poisoned/backdoor data detection, we plot the change of *Attack Success Rate* (ASR), and for noisy feature/label detection, we plot the change of model accuracies after filtering out the low-quality data points selected by each technique.

**I. Backdoor Detection.** Backdoor attacks (Gu et al., 2017) embed an exploit at training time that is subsequently invoked by the presence of a "trigger" at test time. They are considered particularly dangerous since they make victim models predict a target output on inputs with predefined triggers while still retain high accuracy on clean data. Since data points with the backdoor triggers contribute little or negatively to the learning of clean test data, they are expected not to be not included in the subset that optimizes objective (1). This experiment studies the effectiveness of DATASIFTER for removing backdoored examples. We evaluate Trojan attack (Liu et al., 2017) here. We adopted a three-layer CNN as the target model, a poison rate of 0.05, and a target label 'Airplane'. Figure 2 I.(a) and I.(b) elaborate the Trojan attack detection results for a 2,000-size randomly selected subset of the CIFAR-10 dataset. As we can see, DATASIFTER significantly outperforms other approaches; for instance, it achieves a detection rate of 90% with **46%** fewer inspected data points than the others.

**II. Poisoned Data Detection.** In data poisoning attacks, adversaries make slight modifications to some training samples to cause misclassification on target test samples. We evaluate different techniques on feature collision attack (Shafahi et al., 2018) and influence function-based attack (Koh and Liang, 2017). They are clean-label attacks where the attacker does not need to control the labeling of poisoned data. Figure 2 II.(a) and II.(b) show the results for feature collision attack (Shafahi et al., 2018) on a 500-size randomly selected CIFAR-10 subset, where 50 data points of class 'cat' are perturbed with features extracted from a 'frog' sample in the test set. We see that DATASIFTER significantly outperforms all other methods in the poisoned data detection task; for instance, it attains a 90% detection rate with **66%** fewer examined data points.

**III. Noisy Feature Detection.** Noise in features originated from sampling or transmitting (e.g., Gaussian noise) may decrease classification accuracy. Following the settings in Wang et al. (2021a), we add white noise to clean samples, and we evaluate the performance of each technique on detecting those samples. For the CIFAR-10 dataset, we corrupt 25% of the training data images by adding white noise. Based on Figure 2 III.(a) and III.(b), we conclude that DATASIFTER significantly outperforms all other methods on this task; for example, it achieves a 90% of detection rate by examining **67.25%** fewer data points. Meanwhile, the KNN-SV approach exhibits a distinctive trend – it only starts finding the noisy data points until filtering out a certain amount of clean data. This is mainly because all noisy data points are out-of-distribution (OOD). A more detailed discussion is in the Appendix.

**IV. Mislabeling Detection.** Following Ghorbani and Zou (2019), we perform experiments on two datasets and present the results of SVM trained on Enron1 SPAM dataset (Shams and Mercer, 2013). We adopt a bag-of-words representation for training. The noise flipping ratio is 15%. Under this setting, influence-based techniques and G-SV are not applicable since they require the model trained with gradient-based approaches. Figure 2 IV.(a) show that DATASIFTER does not attain the highest detection rate. This is because for SPAM dataset, the margin between the two classes is large so a small amount of mislabeled samples do not significantly affect model performance; hence, those mislabeled samples could evade detection based on the validation performance. Such evasion is acceptable as the model trained over the data points selected by DATASIFTER still achieves a competitive accuracy (as shown in Figure 2 IV.(b)). On the other hand, KNN-SV and LOO can accomplish a good detection rate but end up with a lower model performance. This is because they select very unbalanced samples, as both of them satisfy the symmetry axiom discussed in Section 4.

### 6.2.2 SELECTING HIGH-QUALITY DATA

**V. Data Summarization.** Data summarization aims to select a small, representative subset from a massive dataset, which can retain a comparable utility to that of the whole dataset. We use a CNN trained on the PubFig83 dataset in this experiment. Figure 3 V shows that DATASIFTER and KNN-SV significantly outperform all the other methods.

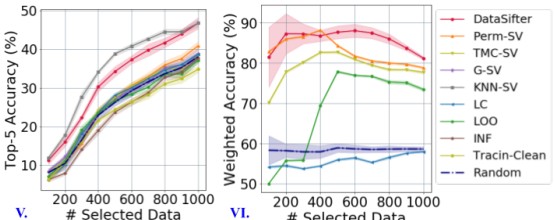

Figure 3: The experimental results for the case of selecting high-quality data (application V and VI). We depict the validation accuracy for both cases. A higher accuracy indicates a better performance.

**VI. Data Debiasing.** We explore whether techniques can help select a subset of training data that improves both fairness and performance for the ML task. We use logistic regression trained on the UCI Adult Census dataset as the task model. We measure the fairness by weighted accuracy – the average of model classification accuracy over females and that over males. G-SV, KNN-SV, and Influence-based techniques are not applicable for this application since they are designed for computing data importance when the metric is test accuracy or loss. Figure 3 VI shows that DATASIFTER achieves the top-tire performance along with the Perm-SV. We note that for this particular task, one can potentially further improve DATASIFTER's performance by changing the subset sampling distribution with more variation in the ratio of sensitive attributes, e.g., gender in Adult dataset. Here, we stick with the uniform subset sampling as other tasks since we intend to evaluate general robustness instead of targeting on one specific data issue type.

## 7 LIMITATIONS AND FUTURE WORK

We propose a principled objective for achieving general robustness in data selection tasks. Based on the objective, we theoretically analyzed the worst-case performance of the existing data valuation-based algorithms and show that these approaches suffer unsatisfying performance guarantees. We present DATASIFTER, a new data selection paradigm guided by the theoretically optimal solution to achieve general robustness against various data issue types, and we showed that DATASIFTER is closer to achieve the goal of general robustness than data valuation-based approaches.

One limitation of our work is the **scalability**. While in Appendix F.5.9, DATASIFTER is often more efficient than other algorithms with similar design goals of general robustness on large datasets, it could still be ineffective for very high dimensional data. Improving the scalability of DATASIFTER through some efficient approximation of data utility functions would be interesting future works.

Another future direction is to explore **other possible architecture or algorithm for utility learning**. While DeepSet-based utility models have shown promising results in our experiment, we can further exploit the approximate submodularity of common data utility functions and use more fine-grained architectures/algorithms for utility learning, e.g., set transformer (Lee et al., 2019) or submodular regularizations (Alieva et al.).

## 8 REPRODUCIBILITY STATEMENT

The anonymized sample implementation for data utility learning and optimization is provided in `https://tinyurl.com/dulwithlabel`. The details of baseline implementations (as well as their source code link, if applicable) are provided in Appendix F.1. The data preprocessing steps are described in Appendix F.2 and detailed in Appendix F.3. The experimented data selection tasks are detailed in Appendix F.3. The details of DeepSets implementations and data utility learning are described in Appendix F.4.

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

## A    PROOF OF THEOREM 1

**Theorem 1.** *For $n \geq 3$, there exists no linear heuristic $\mathcal{M}$ s.t. $d(n, k) = 1$ for every $k \in \{1, \ldots, n\}$.*

*Proof.* Suppose, for contradiction, that there exists a linear heuristic $\mathcal{M}$ s.t. $d(n, k) = 1$ for all $k$. For a dataset $\mathcal{D} = \{1, 2, \ldots, n\}$ and utility function $U$, WLOG assume that the ranks (in non-ascending order) output by $\mathcal{M}$ in the Step 2 of Definition 1 is $(1, \ldots, n)$. Then it means

$$U(\{1\}) \geq U(S) \text{ for all } S \text{ s.t. } |S| = 1,$$
$$U(\{1, 2\}) \geq U(S) \text{ for all } S \text{ s.t. } |S| = 2,$$
$$\ldots$$
$$U(\{1, \ldots, n - 1\}) \geq U(S) \text{ for all } S \text{ s.t. } |S| = n - 1.$$

We construct a simple counter example of $U$ to demonstrate such a $\mathcal{M}$ does not exist: let $n = 3$, we define $U$ as follows:

$$U(\emptyset) = 0,$$
$$U(\{1\}) = 7, U(\{2\}) = U(\{3\}) = 5,$$
$$U(\{1, 2\}) = 9, U(\{1, 3\}) = 9, U(\{2, 3\}) = 10,$$
$$U(\{1, 2, 3\}) = 10.$$

To make $d(3, 1) = 3$, $\mathcal{M}$ must choose 1 for $k = 1$. However, for size-2 subsets, $\mathcal{M}$ can only choose between $\{1, 2\}$ and $\{1, 3\}$, whose utilities are both $9 < U(\{2, 3\})$. Therefore, $d(3, 2) = 2 < \binom{3}{2} = 3$. $\qquad\square$

## B    PROOF OF THEOREM 2

To formally state and prove Theorem 2, we introduce the formal definition of data type here.

**Definition 4.** *Given a dataset $\mathcal{D}$ and utility function $U$, if for all subset $S \subseteq \mathcal{D} \setminus \{i, j\}$, we have*

$$U(S \cup \{i\}) = U(S \cup \{j\}),$$

*we say two data points $i$ and $j$ are of the* same type.

In other words, two data points are of the same type if they will be scored equally by every linear heuristic that satisfies Symmetry Axiom. Theorem 2 essentially says that for all linear heuristic that will assign different scores to different types of data points, their domination ratio can be further upper bounded. We stress that this is a very mild assumption, especially when the space of the scores are continuous, which are the case for most of the existing data value notions.

To simplify the notations for set operations, we use $k \times \{z\}$ to denote a dataset that contains $k$ replicates of data point $z$, and we denote the union of two data sets $S_1 \cup S_2 = S_1 + S_2$. The proof idea of Theorem 2 is to construct a *balanced* dataset that contains same amount of data points from different types. If a linear heuristic $\mathcal{M}$ satisfies symmetry axiom, then $\mathcal{M}$ will select data points of the same type when the target selection number is small, as all data points of the same type will receive the same scores. However, a dataset contains only one type of data points will have nearly no utility.

**Theorem 2** (Restated). *If a linear heuristic $\mathcal{M}$ satisfies symmetry axiom and will always assign different scores for different types of data points, then the domination ratio $d(n, k)$ of $\mathcal{M}$ is upper bounded by $\lfloor n/k \rfloor \left( \binom{\lceil \frac{n}{\lfloor n/k \rfloor} \rceil}{k} \right) / \binom{n}{k}$. In particular, when $c = n/k$ for some constant integer $c$, $d(n, k) \leq \left( \frac{k}{n} \right)^{k-1} = (1/c)^{O(n)}$.*

*Proof.* Suppose there are $c$ *types* of data points: $z_1, \ldots, z_c$. Let $r = n \mod c$. We construct the dataset $\mathcal{D}$ that contains $\lfloor n/c \rfloor$ data points for each of type $z_1, \ldots, z_{n-r}$, and contains $\lceil n/c \rceil$ data points for each of type $z_{n-r+1}, \ldots, z_n$, i.e., $\mathcal{D} = \sum_{i=1}^{n-r} \lfloor n/c \rfloor \times \{z_i\} + \sum_{i=n-r+1}^{n} \lceil n/c \rceil \times \{z_i\}$. We construct utility function $U$ as follows:

$$U(\emptyset) = 0;$$
$$U(i_1 \times \{z_1\} \ldots + i_c \times \{z_c\}) = 1,$$

for every tuple of non-negative integers $(i_1, \ldots, i_c)$ s.t. $1 \leq \sum_{j=1}^{c} i_j \leq n$, except that

$$U(k \times \{z_1\}) = \ldots = U(k \times \{z_c\}) = 0$$

for all $k \leq \lfloor \frac{n}{c} \rfloor$. This construction reflects the rationale that a dataset that only contains one type of data points (e.g., all of the same label) provide little information for training ML models.

Since $\mathcal{M}$ satisfies symmetry axiom, we know that all data points of the same type will receive the same scores. Besides, data points of different types will receive different scores by assumption. Therefore, when the target selection size $k \leq \lfloor \frac{n}{c} \rfloor$, $\mathcal{M}$ will return $k \times \{z_j\}$, which has the worst utilities for subset at size $k$ and there are $(c - r)\binom{\lfloor n/c \rfloor}{k} + r\binom{\lceil n/c \rceil}{k}$ such subsets that only contains single types of data points. For each $k$, by taking the largest possible $c$ such that $k \leq \lfloor \frac{n}{c} \rfloor$, we obtain the desired bound. When $c = n/k$ an integer, we have

$$d(n, k) \leq \frac{n/k}{\binom{n}{k}} \leq \left(\frac{k}{n}\right)^{k-1} = (1/c)^{cn-1} = (1/c)^{O(n)}$$

where the second inequality is due to $\binom{n}{k} \geq (\frac{n}{k})^k$. $\qquad \square$

**Remark 4.** *The upper bound indicates that when $k \leq n/2$, the domination ratio is exponentially small for linear heuristics with symmetry property. We also note the assumption that $\mathcal{M}$ always assigns different scores for different data types can be further relaxed as long as* there exists *such a balanced dataset described in the proof that $\mathcal{M}$ assigns different scores for different data types.*

## C  PROOF OF THEOREM 3

Given a dataset $\mathcal{D} = \{1, \ldots, n\}$ and a submodular utility function $U$, the Shapley value is computed as

$$v_{\text{shap}}(i) = \frac{1}{n} \sum_{S \subseteq \mathcal{D} \backslash \{i\}} \frac{1}{\binom{n-1}{|S|}} \left[ U(S \cup \{i\}) - U(S) \right] \qquad (2)$$

**Theorem 3** (Restated). *The domination ratio $d(n, k)$ of Shapley value-based heuristics is $1/\binom{n}{k}$ for every $n \geq 4$ and any $k \in \{1, \ldots, n\}$, even if we restrict the data utility function $U$ to be submodular.*

*Proof.* We first consider the case when $k \geq 3$.

We construct an instance of a dataset $\mathcal{D} = \{1, \ldots, n\}$ and a submodular utility function $U$ as follows:

$$U(\emptyset) = 0;$$
$$U(\{1\}) = U(\{2\}) = \ldots = U(\{k\}) = 7, U(\{i\}) = 5 \text{ for } i \geq k + 1;$$
$$U(S) = 2|S| + 5 \text{ for all } S \text{ s.t. } 2 \leq |S| \leq k - 1;$$
$$U(\{1, \ldots, k\}) = 2k + 4, \ \ U(S) = 2k + 5 \text{ for all other } S \text{ s.t. } |S| = k;$$
$$U(S) = 2k + 5 \text{ for all } S \text{ s.t. } |S| \geq k + 1.$$

We can compute Shapley value according to its definition in (2):

$$v_{shap}(1) = \ldots = v_{shap}(k) = \frac{1}{n} \left[ 7 + \frac{2(k-1) + 4(n-k)}{n-1} + 2(k-3) + \frac{2\binom{n-1}{k-1} - 1}{\binom{n-1}{k-1}} \right]$$

$$= \frac{1}{n} \left[ 2k + 3 + \frac{4n - 2k - 2}{n-1} - \frac{1}{\binom{n-1}{k-1}} \right]$$

$$v_{shap}(k+1) = \ldots = v_{shap}(n) = \frac{1}{n} \left[ 5 + \frac{2k + 4(n-k-1)}{n-1} + 2(k-3) + \frac{1}{\binom{n-1}{k-1}} \right]$$

$$= \frac{1}{n} \left[ 2k + 1 + \frac{4n - 2k - 4}{n-1} - \frac{1}{\binom{n-1}{k-1}} \right]$$

Since

$$v_{shap}(1) - v_{shap}(k+1) = \frac{1}{n} \left[ 2 + \frac{2}{n-1} - \frac{1}{\binom{n-1}{k-1}} - \frac{1}{\binom{n-1}{k}} \right] \geq \frac{2}{n(n-1)} > 0,$$

we know that $\mathcal{M}$ will always output $\{1, \ldots, k\}$, which achieves the lowest utility among all data subsets of size $k$. Therefore, Shapley value's domination ratio $d(n,k) = 1/\binom{n}{k}$ for all $3 \leq k \leq n-1$.

We then consider the case when $k = 2$. The submodular data utility functions for the case of $k \geq 3$ can be easily adapted as follows:

$$U(\emptyset) = 0;$$
$$U(\{1\}) = U(\{2\}) = 7, U(\{i\}) = 5 \text{ for } i \geq 3;$$
$$U(\{1,2\}) = 8, \ U(S) = 9 \text{ for all other } S \text{ s.t. } |S| = 2;$$
$$U(S) = 9 \text{ for all } S \text{ s.t. } |S| \geq 3.$$

The Shapley value is computed as follows:

$$v_{shap}(1) = v_{shap}(2) = \frac{1}{n} \left[ 7 + \frac{1 + 4(n-2)}{n-1} \right]$$
$$= \frac{1}{n} \left[ 11 - \frac{3}{n-1} \right]$$

$$v_{shap}(3) = \ldots = v_{shap}(n) = \frac{1}{n} \left[ 5 + \frac{4 + 4(n-3)}{n-1} + \frac{2}{(n-1)(n-2)} \right]$$
$$= \frac{1}{n} \left[ 9 - \frac{4}{n-1} + \frac{2}{(n-1)(n-2)} \right]$$

Since

$$v_{shap}(1) - v_{shap}(3) = \frac{1}{n} \left[ 2 + \frac{2}{n-1} - \frac{1}{\binom{n-1}{k-1}} - \frac{1}{\binom{n-1}{k}} \right] \geq \frac{2}{n(n-1)} > 0,$$

we know that $\mathcal{M}$ will always output $\{1, \ldots, 2\}$, which achieves the lowest utility among all data subsets of size 2. Therefore, for Shapley value, $d(n,2) = 1/\binom{n}{2}$.

Finally, we consider the case when $k = 1$. Similarly, we construct a submodular utility function as follows:

$$U(\emptyset) = 0;$$
$$U(\{1\}) = 6, U(\{i\}) = 7 \text{ for } i \geq 2;$$
$$U(\{1,i\}) = 11 \text{ for } i \geq 2, U(\{i,j\}) = 9 \text{ for } i, j \geq 2;$$
$$U(S) = 11 \text{ for all } S \text{ s.t. } |S| \geq 3.$$

The Shapley value is computed as follows:

$$v_{shap}(1) = \frac{1}{n}[6 + 4 + 2] = \frac{12}{n}$$

$$v_{shap}(i) = \frac{1}{n} \left[ 7 + \frac{5 + 2(n-2)}{n-1} + \frac{2(n-2)(n-3)}{(n-1)(n-2)} \right]$$
$$= \frac{1}{n} \left[ 11 - \frac{1}{n-1} \right]$$
$$< \frac{12}{n} = v_{shap}(1).$$

Therefore, Shapley value's domination ratio $d(n,1) = 1/\binom{n}{1}$, which concludes the theorem. $\quad\square$

## D    STOCHASTIC GREEDY ALGORITHM

For completeness, we briefly introduce the stochastic greedy algorithm (SG) for submodular optimization from Mirzasoleiman et al. (2015) here. The stochastic greedy algorithm is a simple approach that, for each iteration, randomly selects a subset of data and then finds the best data point within that subset. In the context of optimizing utility model (i.e., the trained DeepSets model), the "best data point" within each randomly selected subset refers to the data point $z$ with the highest marginal contribution $\hat{U}(S \cup \{z\}) - \hat{U}(S)$, where $S$ is the set of data points selected in previous iterations. The pseudo-code is outlined in Algorithm 1. The distinction between this approach and the vanilla greedy algorithm is that the candidate data points to be selected in each iteration of stochastic greedy algorithm is a smaller, randomly selected subset instead of all unselected data points. Thus, this approach is more efficient than vanilla greedy optimization and the runtime is linear in the number of dataset size to achieve $1 - 1/e - \epsilon$ optimization guarantee for monotone submodular functions ($\epsilon$ is the precision parameter used in Algorithm 1). Although its approximation guarantee has been proved only for monotone submodular objective functions, previous work (Wang et al., 2021a) as well as our experiments show that it also achieve high empirical performance on data utility functions that are approximately submodular, therefore we use this algorithm for optimizing data utility models. Exploring different approaches for optimizing data utility models are interesting future directions.

---

**ALGORITHM 1:** Stochastic Greedy Optimization for Utility Model

---

**input**  : dataset $\mathcal{D}$, trained utility model $\hat{U} : 2^{\mathcal{D}} \to \mathbb{R}$, target selection size $k$, precision parameter for stochastic greedy algorithm $\epsilon$.
**output**: A set $S \subseteq \mathcal{D}$ s.t. $|S| = k$.

1  $S \leftarrow \emptyset$.
2  **for** $t = 1, \ldots, k$ **do**
3      Sample $R \subseteq \mathcal{D} \setminus S$ of size $\frac{|\mathcal{D}|}{k} \log(1/\epsilon)$
4      Find $z = \arg\max_{z \in R} \hat{U}(S \cup \{z\})$
5      $S \leftarrow S \cup \{z\}$
6  **end**
7  **return** $S$

---

## E    CHARACTERIZATION AND LEARNABILITY OF DATA UTILITY FUNCTIONS

**Approximate Submodularity and Data Utility Functions.**    It has been empirically observed in prior works (Wang et al., 2021a) that the data utility functions of many commonly used learning algorithms exhibits approximate submodularity property. Submodular functions are almost always being characterized by the "diminishing return" property. Formally, a set function $f : 2^V \to \mathbb{R}$ returning a real value for any subset $S \subseteq V$ is submodular if $f(\{j\} \cup S) - f(S) \geq f(\{j\} \cup T) - f(T), \forall S \subseteq T, j \in V \setminus T$. Figure 4 shows some examples of the marginal contributions of a *fixed* data point versus the sizes of a sequence of data subsets. We can see a clear "diminishing returns" phenomenon from the figure. That is, an extra training data point contributes less to model accuracy as the base training set size increases.

However, submodularity is an overly stringent condition to characterize data utility functions, especially when the learning algorithm is stochastic. Wang et al. (2021b) propose a relaxation of submodularity condition for modeling common data utility functions as follows:

**Definition 5** ($\beta$-relaxed submodularity (Wang et al., 2021b))**.** *We say that a set function $f$ satisfies $\beta$-relaxed submodularity if for every $S \subseteq T \subseteq N$, and every $j \in N \setminus T$,*

$$f(T \cup \{j\}) - f(T) \leq f(S \cup \{j\}) - f(S) + \beta^{n-(|T|-|S|)} \tag{3}$$

It's been empirically observed that the above condition is satisfied by the data utility functions of commonly used learning algorithms for reasonable values of $\beta$. In particular, when $\beta < 1$, the bias term $\beta^{n-(|T|-|S|)}$ attempts to model the phenomenon that when the datasets are small, the

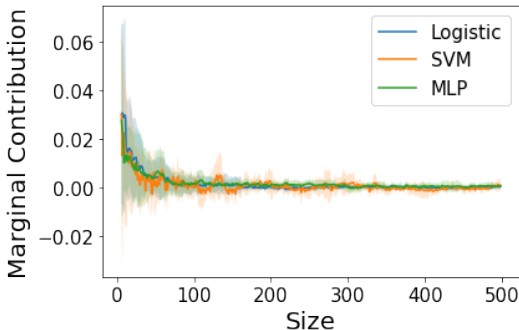

Figure 4: Figure 1 from Wang et al. (2021a): An illustration of "diminishing return" property of data utility functions for widely used learning algorithm (trained on USPS dataset).

contributions of an additional data point to the datasets have larger variance. Hence, when two data sets get more different in sizes, the contributions to two sets might deviate more from the exact submodularity property. When $\beta = 0$, this condition reduces to exact submodularity definition. However, this relaxation is mainly developed based on empirical experiments as well as mathematical convenience. There are no theoretical analysis for the correctness of the above characterization. Although the (approximate) submodularity seems very plausible and intuitive for commonly used data utility functions, the rigorous theoretical analysis is surprisingly hard. Submodularity has thus far been proved for only two simple classes of classifiers–Naive Bayes and Nearest Neighbors (Wei et al., 2015). The rigorous proof of the approximate submodularity of more widely used models such as logistic regression and deep nets seem to be important future works.

**Characterization of Data Utility Functions for Specific Learning Algorithms.** In Section 4, our only assumption of data utility functions is submodularity. Of course, analyzing more restrictive function classes can give tighter results. In fact, a tight analysis of data utility function is possible for simple models (e.g., see the derivation for KNN and Naive Bayes in Wei et al. (2015)). Such analysis seems to be intractable for more complicated models such as deep neural networks. Hence, in Section 4, we tried to find the most restrictive function classes for describing data utility functions without losing generalizability to commonly used ML models. Given the current literature, (approximate) submodularity appears to be the most restrictive function class that can well describe the data utility functions for commonly used ML models. Hence, we devote our analysis in Section 4 to submodular utility functions.

**Theoretical Analysis of the Learnability of Data Utility Functions.** Wang et al. (2021b) also shows that any set functions with range $[0, 1]$ and satisfy the relaxed submodularity condition in Equation 3 belongs to a larger function class called *self-bounding* functions (Boucheron et al., 2000). This class of functions is known for the dimension-free concentration bound and has been proven to be efficiently learnable (Feldman and Vondrák, 2016; Feldman et al., 2020) under mild conditions. This could serve as an insight into why data utility functions can be efficiently learned.

## F    EXPERIMENT DETAILS AND ADDITIONAL RESULTS

### F.1    BASELINE IMPLEMENTATION

The settings of baselines are summarized in Table 3. For fair comparisons between DATASIFTER and baselines, we fix the total number of utility sampling as 4000 for DATASIFTER and all baseline algorithms that require utility sampling, including Perm-SV, TMC-SV, G-SV, and LC. For each sampled subset $S$, we only train model on it once to approximate $U(S)$. Following the settings in Ghorbani and Zou (2019), we set the performance tolerance in TMC-Shapley as $10^{-3}$. Following the

| Baseline | Settings and Hyperparameters |
|---|---|
| Perm-SV | |
| TMC-SV | Tol = $10^{-3}$ |
| G-SV | hyper-parameter search for specific learning algorithm |
| KNN-SV | $K = 5$, use exact calculation formula |
| LC | Break tie by least core vector that has the smallest $\ell_2$ norm |
| LOO | |
| INF | Rank according to influence on the model loss over the clean validation data (the higher, the better); for neural networks, we use stochastic estimation where recursion depth is 5000 and the number of iterations as 10 or larger |
| TracIn-Clean | Rank according to influence on the model loss over the clean validation data (the higher, the better); use exact TracIn described in Pruthi et al. (2020). |
| TracIn-Self | Rank according to influence on the model loss over themselves (the lower, the better) |

Table 3: Summary of Baseline Settings.

settings in Jia et al. (2019c), we set $K = 5$ for KNN-Shapley. We use CVXOPT[1] library to solve the constrained minimization problem in the least core calculation. For influence function technique, we rank training data points according to their influences on the model loss over the validation data. The code is adapted from the PyTorch implementation of influence function on GitHub[2]. For TracIn technique, we use all the parameters in model. We use the exact TracIn instead of the more efficient TracInCP described in Pruthi et al. (2020). That is, we trace the influence of data point $z$ on the loss of $z'$ by sum over all iterations (not checkpoints) in which $z$ is chosen in the batch. TracIn-Self is only used for bad data selection tasks. The intuition is that bad examples are likely to be OOD and have strong influence for themselves. Therefore, when using TracIn-Self to detect bad data, we sort training examples by decreasing self-influence. On the contrary, when using TracIn-Clean, we sort training examples by increasing influence on clean validation data.

For the second-order influence function method from Basu et al. (2020), since the group selection method proposed in the paper is extremely expensive (in addition to the expensive nature of influence function), we do not compare with their method.

### F.2 DETAILS OF DATASETS USED IN SECTION 6

**CIFAR-10 (Krizhevsky et al., 2009).** CIFAR-10 consists of 60,000 3-channel images in 10 classes (airplane, automobile, bird, cat, deer, dog, frog, horse, ship and truck). Each image is of size $32 \times 32$.

**MNIST (LeCun, 1998).** MNIST consists of 70,000 handwritten digits. The images are $28 \times 28$ grayscale pixels.

**Dog vs. Cat (Kaggle).** Dog vs. Cat dataset consists of 2000 images (1000 for 'dog' and 1000 for 'cat') extracted from CIFAR-10 dataset. Each image is of size $32 \times 32$.

**Enron SPAM (Shams and Mercer, 2013).** Enron SPAM dataset consists of 2000 emails extracted from Enron corpus (Klimt and Yang, 2004). The bag-of-words representation has 10714 dimensions, and we perform a $\chi^2$ test to the dataset to retrieve only $10\%$ best features (so the feature dimension becomes 1071).

**PubFig83 (Pinto et al., 2011).** PubFig83 is a real-life dataset of 13,837 facial images for 83 celebrities, where the images are of varying quality, e.g., some images contain two human faces. We resize each image to $32 \times 32$.

**Covid-CT (Zhao et al., 2020).** The COVID-CT-Dataset has 746 CT images in total, containing 349 images from 216 COVID-19 patients and the rest of them are from healthy people. The dataset is separated into 543 training images and 203 test images. We resized each image to $32 \times 32$.

---

[1] https://cvxopt.org/
[2] https://github.com/nimarb/pytorch_influence_functions

| Data Quality Issue | Dataset | Size | Bad Data Rate | Relevant Hyperparameters |
|---|---|---|---|---|
| Backdoor - Trojan Attack | CIFAR-10 | 2000 | 0.05 | Square trigger on bottom-right corner |
| Backdoor - BadNets | MNIST | 1000 | 0.25 | Pattern Backdoor on bottom-right corner |
| Poisoning - Feature Collision | CIFAR-10 | 500 | 0.1 | $\beta = 0.25$, learning rate $500 \times 255$, 120 iterations |
| Poisoning - Influence-based | Dog vs Cat | 2000 | 0.025 | $\alpha = 0.02$, 100 iterations |
| Noisy Feature | CIFAR-10 | 2000 | 0.25 | coordinate-wise Gaussian noise with $\sigma = 1$ |
| Noisy Feature | MNIST | 1000 | 0.25 | coordinate-wise Gaussian noise with $\sigma = 1$ |
| Mislabeling | SPAM | 500 | 0.15 | Flip label uniformly at random |
| Mislabeling | CIFAR-10 | 500 | 0.25 | Flip label uniformly at random |

Table 4: Summary of Hyperparameters in Bad Data Detection tasks. For Feature collision data poisoning attack, we followed public source code `https://github.com/Trusted-AI/adversarial-robustness-toolbox/blob/main/art/attacks/poisoning/feature_collision_attack.py` to apply the poison frogs attack (feature collision attack). The implementation does not hardcode the dynamical calculation of the dimension of the feature layer, thus requiring a larger learning rate to obtain similar coefficients. We fine-tuned the model with a learning rate ($\lambda$) of 500*255.0 and obtained the successful attack.

**UCI Adult Census (Dua and Graff, 2017).** The Adult dataset contains 48,842 records from the 1994 Census database. Each record has 14 attributes, including the sensitive gender and race information. The task is to predict whether one's income exceeds \$50K/yr based on census data. The categorical attributes in the dataset are one-hot encoded.

**COMPAS (Yoon, 2018).** We use a subset of the COMPAS dataset that contains 6172 data records used by the COMPAS algorithm in scoring defendants, along with their outcomes within two years of the decision, for criminal defendants in Broward County, Florida. Each data record has features including the number of priors, age, race, etc. The categorical attributes in the dataset are one-hot encoded.

### F.3 EXPERIMENTS IMPLEMENTATION

The attack success rate of backdoor attacks is measured by a separate test backdoored dataset. For the experiment on backdoor detection, data poisoning detection, noisy detection, and mislabel detection on the CIFAR-10 dataset, the CNN model we use has two convolutional layers. A max-pooling layer follows each with the ReLU as the activation function, and followed by three fully-connected layers. For the experiment on Backdoor detection and noisy feature detection on the MNIST dataset, we use LeNet adapted from LeCun et al. (1998), which has two convolutional layers, two max-pooling layers, and one fully-connected layer. For the experiment on data summarization on PubFig83, we use a simplified VGG architecture[3]. For the experiment on poisoning detection over the Dog vs. Cat dataset as well as the data summarization over the COVID-CT dataset, we use a small CNN model adapted from PyTorch tutorial[4]. We use Adam optimizer with learning rate $10^{-3}$, mini-batch size 32 to train all of the models mentioned above for 30 epochs, except that we train LeNet for 5 epochs on MNIST. For the experiment on data biasing on the Adult dataset, we implement logistic regression in scikit-learn (Pedregosa et al., 2011) and use the LibLinear solver. For the experiment on mislabeling detection on SPAM and data debiasing on COMPAS, we adopt SVM implementation from scikit-learn library (Pedregosa et al., 2011) with RBF kernel. The $\chi^2$ test used for pre-processing Enron SPAM dataset is implement by scikit-learn library for univariate feature selection.

### F.4 DATA UTILITY LEARNING CONFIGURATION

A DeepSets model is a set function $f(S) = \rho \left( \sum_{x \in S} \phi(x) \right)$ where both $\rho$ and $\phi$ are neural networks. Since summation does not depend on the permutation of elements in $S$, the architecture is permutation-invariant and thus a set function. In our experiment, both $\phi$ and $\rho$ networks have three fully-connected layers. We note that $\phi$ network can also be convolutional for image datasets. However, we find that for image datasets CIFAR-10, Dog vs Cat, PubFig83 and Covid-CT, it is better to use the data features extracted by the convolutional layers of the target model (trained on the full training set) as the input for DeepSets models. All labels are one-hot encoded, and each label vector $y_i$ is concatenated with

---

[3]`https://github.com/YiZeng623/frequency-backdoor`
[4]`https://pytorch.org/tutorials/beginner/blitz/cifar10_tutorial.html`

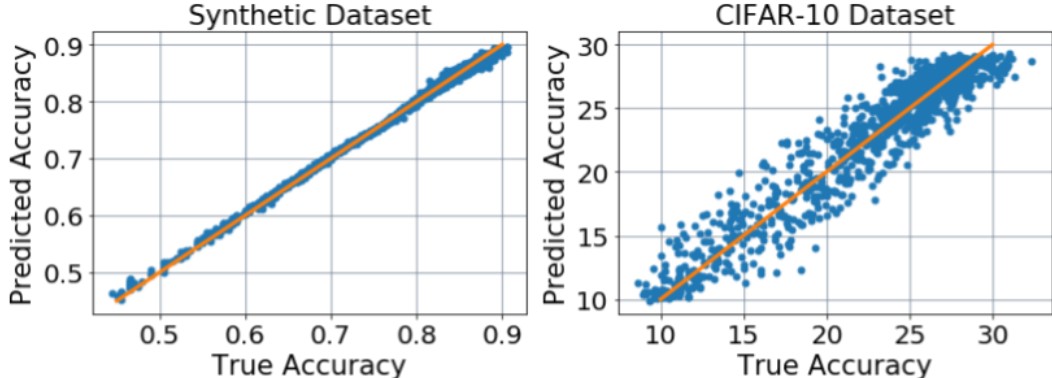

Figure 5: Predicted vs. True Utility for unseen subsets of (a) logistic regression classifier trained on a synthetic dataset, and (b) CNN model trained on a subset of CIFAR-10 dataset.

the corresponding data feature vector $x_i$, and then feed $(x_i, y_i)$ into the DeepSets model as the input. We tune the DeepSets model architecture for different datasets. For the COMPAS dataset, we set the number of neurons in every hidden layer and the dimension of set features (i.e., the output of $\phi$ network) to be 64. For all other datasets, we set the number of neurons and set dimension to be 128. We use the Adam optimizer with learning rate $10^{-4}$, mini-batch size of 32, $\beta_1 = 0.9$, and $\beta_2 = 0.999$ to train all of the DeepSets utility models, for up to 20 epochs. The number of utility samples we use is 4000 for all experiments, as mentioned in Section 5. To sample a subset, we first uniformly pick a set size, and then uniformly sample a subset of the given size without replacement. We left exploring the relationship between subset sampling distribution and data selection performance as our future work. The validation data in utility sampling are 500 clean data points sampled from the test data of the corresponding datasets.

## F.5   ADDITIONAL RESULTS

### F.5.1   GENERALIZATION OF DATA UTILITY LEARNING

In Figure 5, we show examples of correlation plot between predicted and actual data subset utility. For Figure 5 (a), the data utility function is the validation accuracy of logistic regression model trained on a synthetic dataset. For the synthetic data generation, we sample 200 data points from a 50-dimensional Gaussian distribution, where the covariance matrix is identity matrix and the 50 mean parameters are sampled uniformly from $[-1, 1]$. Each data point is labeled by the sign of the sum of the data point vector. Since logistic regression problem is convex and has unique global minimum, the data utility has smaller variance due to the randomness in learning algorithm and thus utility prediction is relatively accurate. For Figure 5 (b), the data utility function is the validation accuracy of a CNN model trained on a subset of CIFAR-10 dataset of size 2000. Since learning neural networks is a non-convex problem, the variance in data utility from the optimization process is relatively large, the utility prediction error is also relatively large. However, we can still need a strongly positive correlation between the predicted and true data utility, which means that the data utility model is still able to differentiate between subsets with large utility gaps.

### F.5.2   EFFECT OF SYMMETRY AXIOM

To better illustrate the issue raised by symmetry axiom (Section 4), we evaluate data selection performance of LOO, Shapley, and least core heuristic on a synthetic dataset with 15 training data points (so that we can compute the exact Shapley and least core values, as well as obtain the optimal subset). The tiny synthetic dataset is generated by sample data points from a 2-dimensional standard Gaussian distribution, where the mean vector of the Gaussian distribution is $(0.1, -0.1)$. Each data point is labeled by the sign of its vector's sum. We first sample 9 data points with positive label and 2 data points with negative label. We then replicate each of the two negatively labeled data points for two times. To simulate natural noise, we add Gaussian noise to the copied data vector with scale $10^{-5}$. By sampling and copying, we obtained 15 data points with natural redundancy. The utility

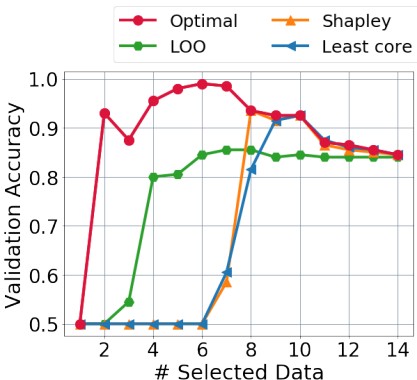

Figure 6: Results of data selection with different heuristics on a tiny dataset with natural redundancy. Dataset and implementation are detailed in the Appendix.

metric is the test accuracy of a *Support Vector Machine* (SVM) classifier trained on the dataset. Since there are only 6 data points with negative label, they tend to be assigned with larger (and similar) importance scores by linear heuristics like Shapley value. Both Shapley and Least core thus rank negative points with higher importance. This means that when the target selection size is less than 6, the selected dataset contains only data of negative label and no information about the positive label class at all. As shown in Figure 6, both Shapley and Least core achieves trivial utility for the first 6 selected data points. With small selection budgets, the subsets selected by all the heuristics have low utility as the heuristics fail to promote diversity during selection due to symmetry axiom, and thus suffer poor data selection performance.

### F.5.3 BACKDOOR ATTACK

We consider the two most popular types of backdoor attacks, namely the BadNets (Gu et al., 2017) and the Trojan square trigger (Liu et al., 2017). Those two attacks' major difference is the trigger itself, where BadNets adopts a white block trigger at the right corner, and Trojan attack adopts a square trigger.

Here, we show the results of DATASIFTER and baseline techniques over detecting BadNets triggers on MNIST dataset. The poisoning rate is 0.25, and the target label is '0'. The performance of different techniques is illustrated in Figure 7 I.(a) and I.(b). We can see that DATASIFTER outperforms all other methods in the detection rate and significantly reduces the attack accuracy after filtering out bad data points. Besides, we can see that TracIn-Clean and influence function exhibits similar trends, which only starts finding the adversarially perturbed data points until filtering out a certain amount of clean data (same thing happen for data poisoning detection). We conjecture that this is because adversarially perturbed data points are designed to change the model prediction of particular target examples; each individual perturbed data point, however, does not have a significant impact on the loss of clean validation data. Besides, adversarially perturbed data points are similar and their influence on validation data is also similar in nature, thus they tend to be ranked together by influence-based methods.

### F.5.4 DATA POISONING ATTACK

We discuss two popular types of clean-label data poisoning attacks. Feature collision attack (Shafahi et al., 2018) crafts poison images that collide with a target image in feature space, thus making it difficult for a model to discriminate between the two. Influence function-based poisoning attack (Koh and Liang, 2017) identifies the most influential training data points for the target image and generates the adversarial training perturbation that causes the most increase in the loss on the target image. The Attack Success Rate is measured by the model's confidence on the prediction of poisoned data (with respect to the target label).

Figure 7 II.(a) (b) show the results for influence function-based attack on Dog vs. Cat dataset, where 50 data points of class 'cat' are perturbed to increase the model loss on a 'dog' sample in the test

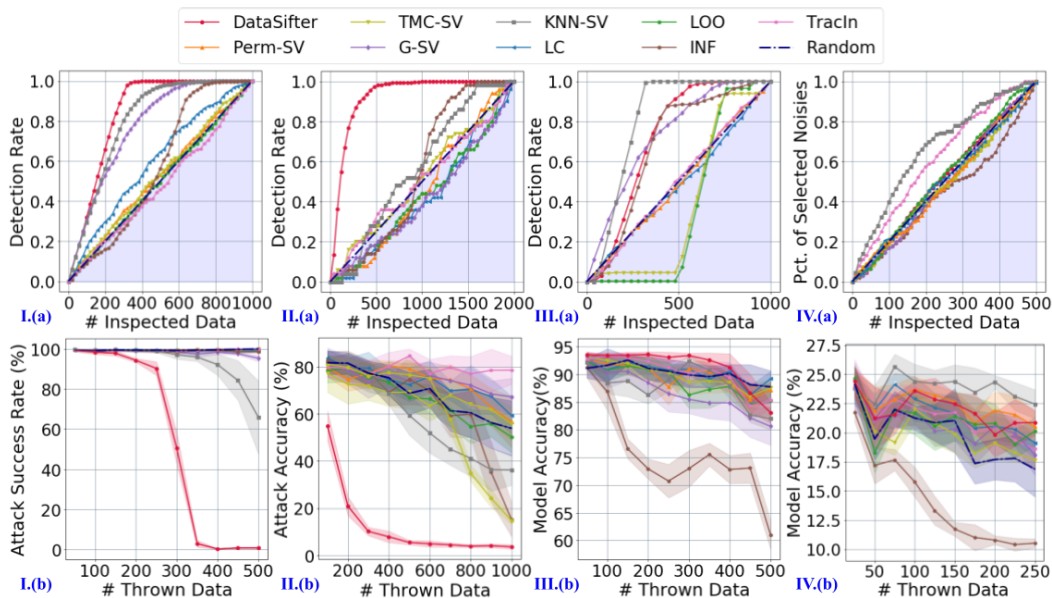

Figure 7: The experimental results and comparisons of the DATASIFTER under the case of filtering out harmful data (application I-IV). The light blue region in each (a) graph represents the area that a method is no better than a random selection. For I.(b) and II.(b), we depict the Attack Success Rate (ASR), where a lower ASR indicates a more effective detection. For III.(b) and IV.(b), we show the model test accuracy, where a higher accuracy means a better selection.

set. As we can see, DATASIFTER is a more effective approach to detect poisoned data points than all other baselines.

### F.5.5 NOISY FEATURE

We follow the same evaluation method for noisy data detection as in Section 6 with another setting: LeNet model trained on noise polluted MNIST. We randomly select 1000 data points and corrupt 25% of them with white noise. As shown in Figure 7 III.(a) (b), we can see that although KNN-Shapley can achieve slightly better performance in detecting noisy data points, DATASIFTER still retains a higher performance for model accuracy. Besides, similar to the case for CIFAR10, we find that the KNN-SV approach only starts finding the noisy data points until filtering out a certain amount of clean data. This is mainly because all noisy data points are out-of-distribution (OOD), as shown in Figure 8 (b). The mechanism of KNN-SV, however, tends to assign 0 values to OOD data points while assign negative values to clean data points that are in-distribution but have different labels from their neighbors. Figure 8 (c) gives a visualization of the distribution of KNN-Shapley values.

### F.5.6 MISLABELED DATA

We conduct another experiment on noisy label detection: a small CNN model trained on 500 data points from the CIFAR-10 dataset. The noise flipping ratio is 25%. The performance of mislabel detection is shown in Figure 7 IV.(a). As we can see, no techniques are particularly effective in detecting mislabeled data for this task. Only KNN-SV and TracIn-Self achieves a slightly better performance than other approaches. We conjecture that the difficulty of mislabel detection on CIFAR-10 dataset is due to the following reason: since an oracle for detecting mislabeled data points can also be used to implement a classifier, the difficulty of mislabeling detection is at least as difficult as classification. A classifier directly trained on the 500 clean data points in this experiment, however, can only attain around 28% test classification accuracy. Nevertheless, Figure 7 IV.(b) shows that DATASIFTER only achieves slightly worse model accuracy than KNN-SV after filtering out selected bad data points. Besides, we can see that although TracIn-Self is able to filter out more mislabeled data points than others, it does not lead to higher accuracy for the selected data as TracIn-Self ignores data interaction and end up selecting similar data points.

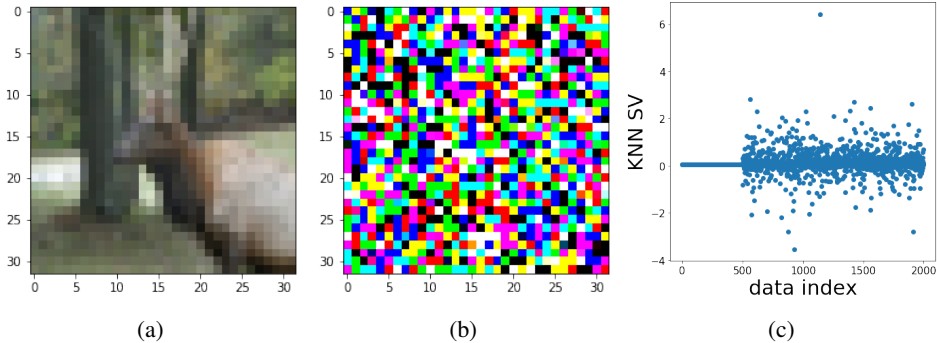

Figure 8: (a) a normal image from CIFAR-10, (b) an example of noisy data image, (c) a sample of KNN-Shapley values, where data points with index $< 500$ are noisy. A data point with a higher KNN-Shapley value is considered more important.

### F.5.7 DATA SUMMARIZATION

As another setting for the data summarization application we consider, we use the patient CT images from COVID-CT dataset for a binary classification task, which aims to determine whether an individual is diagnosed with COVID-19 or not. The CNN model trained on the dataset achieves around 72% classification accuracy. Figure 9 V. shows the results for selecting up to 400 data points with different

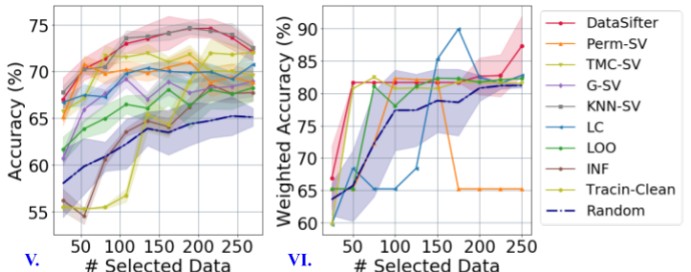

Figure 9: The experimental results and comparison of the DATASIFTER under the case of selecting high-quality data (application V and VI). We depict the validation accuracy for both cases. A higher accuracy indicates a better performance.

techniques. As we can see, DATASIFTER achieves the best model accuracies on the selected data points along with KNN-SV.

### F.5.8 DATA DEBIASING

We introduce another data debiasing experiment on the criminal recidivism prediction (COMPAS) task, where races are considered as the sensitive attribute. The utility metric we adopted here is the average accuracy across different race groups. The learning algorithm we use is SVM with RBF kernel. Baselines including G-SV, KNN-SV, and Influence-based techniques are not applicable for this application due to the utility metric and learning algorithm we use. Figure 9 VI. shows the results for DATASIFTER and the remaining five baselines. We can see that DATASIFTER again achieves the top-tire performance.

### F.5.9 UTILITY OPTIMIZATION ON LARGER DATASETS

We compare the scalability between DATASIFTER and other baselines on large datasets. We show the results for backdoor detection on a 10,000-size Trojan square poisoned CIFAR-10 dataset here. For DATASIFTER, we only sample data subset utilities from 1000 data points (10% of the whole dataset), while the learned utility model (the DeepSets model) is optimized over the entire 10,000 data points. The intuition of this approach is that the learned utility model can provide utility estimations for sets of seen data points as well as unseen data points. Therefore, the greedy optimization can be performed on a larger dataset even if the utility learning is on smaller subsets. Since the learned utility model may not generalize well on dataset of sizes greater than 1000, we start over the optimization process once the selected data points exceed 1000, and then select another 1000

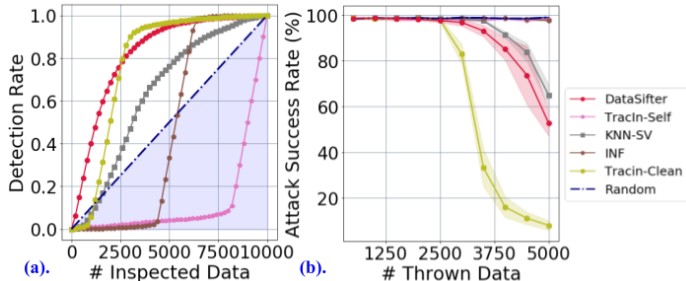

Figure 10: The experimental results and comparison of the DATASIFTER and baseline algorithms for detecting backdoored data on larger datasets.

data points until we reach the target selection budget. The pseudo-code is summarized in Algorithm 2.

---

**ALGORITHM 2:** Utility Optimization on Large Datasets

---

**input** : dataset $\mathcal{D}$, trained utility model $\hat{U}$, target selection size $k$, the maximum size of sets where $\hat{U}$ has seen during utility training $B$ (usually much smaller than $\mathcal{D}$), precision parameter for stochastic greedy algorithm $\epsilon$.

**output :** A set $S \subseteq \mathcal{D}$ s.t. $|S| = k$.

8  $S \leftarrow \emptyset$, $S_0 \leftarrow \emptyset$.

9  **for** $t = 1, \ldots, k$ **do**

10    Sample $R \subseteq \mathcal{D} \setminus S$ of size $\frac{|\mathcal{D}|}{k} \log(1/\epsilon)$.

11    Find $z = \arg\max_{z \in R} \hat{U}(S \cup \{z\})$.

12    $S_0 \leftarrow S_0 \cup \{z\}$.

13    **if** $|S_0| = B$ **then**

14      $S = S \cup S_0$.

15      $S_0 \leftarrow \emptyset$.

16    **end**

17    $S = S \cup S_0$.

18 **end**

19 **return** $S$

---

When executed on NVIDIA Tesla K80 GPU, the clock time for the utility sampling step is within 15 hours for 4000 utility samples with a small CNN model, as the data size is fairly small. The LOO, the Least core, and all the Shapley value-based approaches except KNN-SV did not terminate in 24 hours, so we remove them from comparison. As we can see from Figure 10, DATASIFTER once again outperforms all the remaining approaches. The results show that the learned utility model can provide utility estimations for sets of unseen data points, which largely improves the scalability of DATASIFTER. On the contrary, the existing valuation-based approaches cannot predict the importance of unseen data points. Thus their utility sampling has to be conducted over the entire dataset.

Follow the same method in Algorithm 2, we perform noisy data detection on a 20,000-size CIFAR-10 dataset. The corruption ratio is 25%. Again, for DATASIFTER, we train the utility model on utility samples collected from retraining on subsets of 1000 data points (5% of 20,000). We remove the LOO, the Least core, and all the Shapley value-based approaches except KNN-SV from comparison, as they did not terminate in 24 hours for 4000 utility sampling on the 20,000-size set. As we can see from Figure 11 (a) and (b), DATASIFTER significantly outperforms all other baseline techniques except TracIn-Clean. However, TracIn-Clean only starts finding bad data after filtering out certain clean data points. The results demonstrate that although the utility sampling step could be expensive, the scalability of DATASIFTER can be boosted by the predictive power of the learned utility model.

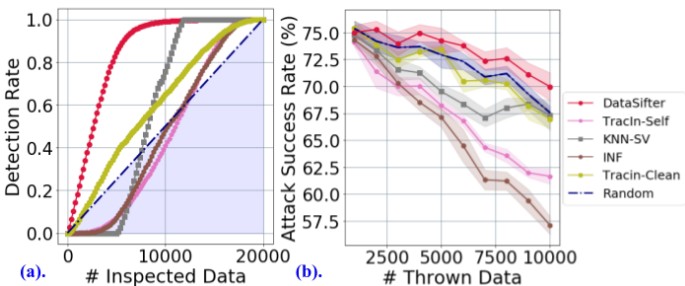

Figure 11: The experimental results and comparison of the DATASIFTER and baseline algorithms for detecting noisy data on larger datasets.

