# OpenReview forum: "Towards General Robustness to Bad Training Data"
_ICLR.cc/2022/Conference — ICLR 2022 Submitted_

### Official Review · Reviewer_s9Xw · 2021-11-01

**Correctness:** 3
**Technical Novelty And Significance:** 3
**Empirical Novelty And Significance:** 2
**Recommendation:** 6
**Confidence:** 3

**Main Review:**

\textbf{Strength:}

1. This paper suggests an algorithmic framework based on utility optimization approach to achieve robustness against general bad data. The method performs well empirically.

2. Theoretically shows that Shapley value based methods has small domination ratio.

\textbf{Weakness:}

1. Regarding Theorem 3: "...even if the data utility function U is submodular"  Bad worst case performance could be due to restricting U to be submodular. What happens if U is not submodular? I disagress with the choice of words "even if". In my opinion it should read "...when the data utility function U is submodular."

2. It should be highlighted clearly in the main paper (not in appendix) that the better empirical performance is for a specific choice of utility function $u$, ,e.g., average accuracy in debiasing. Since choice of utility function is the most important part of detecting bad data, and the supremacy of one method over the other may not hold for a different utility function.

3. The authors criticize linear heuristic methods showing that worst case domination ratio of the methods are small. But there is no theoretical analysis of DataSifter. It performs only empirically better, and since the experiments are done for a specific choice of utility function, nothing can be said about its worst case performance, and it could as well be as small as linear heuristics.

4. The DeepSets approach to utility optimization has been introduced by Wang et.al. 2021a (as the authors agree) and this method itself is not a novel contribution of this work (including the label information does not contribute anything new to the idea).

\textbf{Minor issues:}

1. On page 2: "Guided by goals of downstream..." It is not clear to me what this line means.

2. Line 7, page 5: adds ----> add

**Summary Of The Paper:**

In this work, the authors:

1. shows that linear heuristic like Shapley value based methods do no perform well w.r.t worst case performance in selecting the best subset of data.

2. develops an algorithmic framework based on utility optimization approach to achieve robust ness against general \textit{bad} data

**Summary Of The Review:**

My decision is borderline accept for this paper only because of empirical success in some specific examples and theoretical results on the domination ratio of Shapley value based methods. Although, I should mention this paper has considerable weakness. Depending on discussion with other reviewers I may increase/decrease my score.

---

> ### Author Response · Authors · 2021-11-19
> **Response to Reviewer s9Xw**
>
> **Q. [Bad worst-case performance could be due to restricting $U$ to be submodular?]**
>
> **A.** We respectfully disagree with the argument that ``bad worst-case performance could be due to restricting U to be submodular.” The worst-case performance of arbitrary utility functions is actually a lower bound of the worst-case performance of submodular utility functions. As submodular utility functions are subsumed by general utility functions, minimizing the performance over submodular utility functions (a more restricted set) should result in a higher performance value than minimizing the performance over general functions (a large set).
>
> **Q. [Highlight the choice of Utility function]** *“It should be highlighted clearly in the main paper (not in appendix) that the better empirical performance is for a specific choice of utility function $u$,e.g., average accuracy in debiasing.”*
>
> **A.** We have highlighted the choice of utility functions in the main text. We agree that the choice of utility functions is important to detect bad data. In fact, the specification of utility functions depends on the goal of data selection/removal and should be done in a context-dependent manner. With that being said, the specification of data utility is often straightforward. For instance, detecting poisoning attacks, backdoor attacks, mislabeled data, noisy features, and redundant data share the same goal of improving model accuracy; hence, a natural utility function for all these tasks is just model accuracy. In our evaluation, only the debiasing task requires a different utility function (the weighted accuracy across different demographical groups).
>
>
> **Q. [Clarification of Task-driven]** *“On page 2, what means “Guided by goals of downstream ...?”*
>
> **A.** We have updated the main text and hope it clarifies the confusion. One of the major advantages of DataSifter is the “task-driven” property. Here, “task” is referred to as a general ML task, such as training a classifier/regressor to have high utility (e.g., accuracy, fairness, etc.); “driven” refers to the process where we obtain the data-utility pairs (i.e., utility samples) from the trained classifier/regressor. Therefore, by “task-driven”, we refer to the fact that our method directly uses the utility samples collected from the downstream ML task for utility learning. For example, if our goal is to train a classifier that is both highly accurate and fair (by measures like equalized odds), then we will train multiple classifiers (based on different subsampled training datasets), each of which will provide us a valid utility sample point. By learning from the collection of these utility samples, we adapt our utility function to this specific task.
>
> **Q. [Theoretical Analysis of DataSifter?]** *"The authors criticize linear heuristic methods showing that worst-case domination ratio of the methods is small. But there is no theoretical analysis of DataSifter."*
>
> **A.** We apologize for not being as clear as we intended. As the DataSifter optimizes the data utility function (under the same unbounded computation assumption made for the analysis of linear heuristics), the domination ratio is just 1. In other words, the worst-case performance of DataSifter is optimal. We have clarified the worst-case analysis for DataSifter at the end of the first paragraph of Section 5.
>
>
> **Q. [Does the choice of utility function affect the validity of worst-case performance comparison?]** *“Since the experiments are done for a specific choice of utility function, nothing can be said about its worst-case performance, and it could as well be as small as linear heuristics.”*
>
> **A.** We should have clarified that both DataSifter and other data valuation-based baselines (i.e., linear heuristics) share the same utility function in experiments. Hence, the choice of utility function does not affect the comparison of worst-case performance between DataSifter and the baselines. We have highlighted this point at the end of the first paragraph of Section 6.1.

---

> > ### Author Response · Authors · 2021-11-27
> > **Message to Reviewer s9Xw**
> >
> > Dear Reviewer s9Xw,
> >
> > We'd like to express our gratitude once more for your constructive suggestions, which resulted in interesting revision updates. We've responded to each of your questions. Hopefully, you'll find that they adequately address your concerns. Additionally, we'd like to know if you have any additional questions or require clarification before the rebuttal phase concludes. We would be delighted to address them in the revision.
> >
> > Best wishes,
> >
> > Authors of Paper

---

> > ### Author Response · Authors · 2021-12-03
> > **Follow-up Message to Reviewer s9Xw**
> >
> > Dear Reviewer,
> >
> > We are wondering whether your concerns have been properly addressed. If you have further questions after reading our responses, it would be great to let us know.
> >
> > Best regards,
> >
> > The authors.

---

### Official Review · Reviewer_4LgM · 2021-11-02

**Correctness:** 3
**Technical Novelty And Significance:** 2
**Empirical Novelty And Significance:** 2
**Recommendation:** 5
**Confidence:** 3

**Main Review:**

The paper is well-written and easy to follow. The problem of identifying harmful or bad training data is interesting and important in dataset creation and cleaning. The theory for worst case performances of valuation based approaches is motivating but it doesn't necessarily imply the method will fail as it's only a worst case analysis. The other drawback is that the DataSifter algorithm is highly empirical with no theoretical support. Some detailed comments are made as follows.

1. The counterexamples in Theorem 1 to 3 are highly constructive in $U$ and might not reflect actual performances of linear heuristics on real datasets. Especially if the utility function has certain structure and cannot be defined arbitrarily. The experiment in E.5.1 uses SVM as utility metric which as a special instance can be much better than the worst-case performance bound (and the same is also true for a wider range of utility functions used in practice). I would like results more with additional constraints on the utility function or lower bounds for a particular class of utility functions used in practice.
2. The paper shows good empirical performance of the DataSifter algorithm in certain identification tasks. It seems a bit unrelated to the first part of the paper as the good performance for some datasets doesn't imply a higher worst-case lower bound. A more reasonable comparison is to either provide a better worst-case lower bound for DataSifter (maybe under assumptions like exact recovery $\widehat{U} = U$) or as pointed out in my first point to give instance-specific performance upper bound for the valuation-based approaches.

In all, I think the paper gives some interesting empirical results but the theory doesn't exactly corroborate the main argument that we should prefer DataSifter over coventional heuristics. I incline to reject the paper.

**Summary Of The Paper:**

In this paper, the authors study the problem of data selection in the presence of bad training data by maximizing the utility function $U$ over all subsets of cardinality $k$. The authors first show that the valuation based approaches can have bad performances, and propose the DataSifter algorithm to approximately solve the combinatorial optimization problem.

**Summary Of The Review:**

The paper provides worst-case analysis of data selection using linear heuristics, revealing potentially hazards using valuation-based approaches, which is interesting and novel. However, it doesn't exactly corroborate the main argument that we should prefer DataSifter over coventional heuristics.

---

> ### Author Response · Authors · 2021-11-19
> **Response to Reviewer 4LgM**
>
>
> **Q. [Worst-case analysis when there are additional constraints on utility functions]** *“I would like results more with additional constraints on the utility function or lower bounds for a particular class of utility functions used in practice.”*
>
> **A.** Thanks for this great suggestion! We concur with the reviewer that analyzing more restrictive function classes can give tighter results. In fact, a tight analysis of data utility function is possible for simple models (e.g., see the derivation for KNN and Naive Bayes in [WIB15]). However, such analysis is not tractable for more complicated models such as deep neural networks. Hence, we tried to find the most restrictive function classes for describing data utility functions *without losing generalizability to commonly used ML models*. Given the current literature, (approximate) submodularity appears to be the most restrictive function class that can well describe the data utility functions for commonly used ML models. Hence, we devote our analysis to submodular utility functions.
>
> [WIB15] Submodularity in Data Subset Selection and Active Learning, ICML’15
>
> **Q. [Worst-case lower bound for DataSifter under the assumption of $\hat U = U$?]** *”A more reasonable comparison is to either provide a better worst-case lower bound for DataSifter (maybe under assumptions like exact recovery $\hat U = U$) or to give instance-specific performance upper bound for the valuation-based approaches.”*
>
> **A.** We apologize for not being as clear as we intended. As the DataSifter optimizes the data utility function (under the same unbounded computation assumption made for the analysis of linear heuristics), the domination ratio is just 1. In other words, the worst-case performance of DataSifter is optimal. We have clarified the worst-case analysis for DataSifter at the end of the first paragraph of Section 5.

---

> > ### Author Response · Authors · 2021-11-27
> > **Message to Reviewer 4LgM**
> >
> > Dear Reviewer 4LgM,
> >
> > We'd like to express our gratitude once more for your constructive suggestions, which resulted in interesting revision updates. We've responded to each of your questions. Hopefully, you'll find that they adequately address your concerns. Additionally, we'd like to know if you have any additional questions or require clarification before the rebuttal phase concludes. We would be delighted to address them in the revision.
> >
> > Best wishes,
> >
> > Authors of Paper

---

> > ### Author Response · Authors · 2021-12-03
> > **Follow-up Message to Reviewer 4LgM**
> >
> > Dear Reviewer,
> >
> > We are wondering whether your concerns have been properly addressed. If you have further questions after reading our responses, it would be great to let us know.
> >
> > Best regards,
> >
> > The authors.

---

### Official Review · Reviewer_yuTL · 2021-11-02

**Correctness:** 3
**Technical Novelty And Significance:** 2
**Empirical Novelty And Significance:** 2
**Recommendation:** 6
**Confidence:** 4

**Main Review:**

Overall, I enjoyed reading the paper.  It was well-structured, easy to follow, and presented ideas clearly.  Nonetheless, it is my view that the paper still needs more work in particular around the experimental evaluation.

**Strengths**:

1) The idea of training empirical estimator $\hat{U}$ is simple (a strength) and nice.

2) Most recently published methods too narrowly target specific forms of “bad data”, specific learning domains (e.g., vision), specific model types, or even specific adversarial attacks.  More work needs to be directed towards agnostic robustness, which the authors do here.  Such general-purpose methods are generally harder to achieve (both theoretically and empirically) and are more useful in practice.

3) Your empirical evaluation includes an impressively large number of baselines.  This level of thoroughness is commendable.

**Weaknesses**:

1) The proposed methodology relies on utility metrics displaying "'approximate' submodularity."  Intuitively such submodularity is very plausible.  However, more detail is needed.  A more quantitative understanding/definition of what constitutes "approximate" would be appreciated.  Similarly, reading the first paragraph of Sec. 5 seems to imply that while many potential applications exhibit "'approximate' submodularity", others do not. Even if it only appears in the supplement, expanding this discussion so a potential reader knows when DataSifter applies (and more importantly when it does not) is important.  It is also unclear how the submodularity approximation affects DataSifter’s practical worst case performance.

2) Beyond the number of baselines, I have concerns about other aspects of your experimental design.

* I would have appreciated seeing empirically how well $\hat{U}$ performs in practice (e.g., a correlation plot between predicted and actual subset utility perhaps across different size subsets).  This could be in the supplement.

* Reviewing your supplemental materials, most of the models you consider are small in size (please correct me if I am wrong).  Do you have any analysis on large models -- e.g., SOTA, modern transformers?  Even if numerical values cannot be included now, can you anecdotally speak to how well your method has performed on such models (both in terms of execution time and numerical performance)?

* Mislabeling Detection (6.2.1.IV) - It seems that DataSifter underperforms some Shapley Value-based methods for moderately sized datasets.  I did not find the explanation around small amounts of data not affecting validation accuracy very strong.  While in absolute terms, the change in accuracy is small, the relative change in error rate seems large.  Perhaps I am missing something.

* For the experiments, I would have preferred a table in the supplemental materials that summarized the various dataset sizes, attack set sizes, perturbation distances etc.  It was time consuming to piece that various information together from reading the individual paragraphs.

* If I understood Sec. 6.2.1 and suppl. Sec. E.5.2 correctly, the backdoor experiments used a poisoning rate of 20%-25%, which is high and somewhat unrealistic in practice.  Also, by having a high poisoning rate, influence based methods generally will perform poorly since the effect of the attack is diffused over all the attack samples.  How does your method perform with much smaller poisoning rates?

* Methods like TracIn and Influence Functions consider a specific target example.  Are the attack success rate values w.r.t. to the specified target or any backdoored sample?  How was the target selected?

* In suppl. Section E.3, your proposed model for CIFAR10 has two convolutional layers and three fully-connected layers.  This seems a peculiar choice for CIFAR10 classification and probably would not perform well on larger dataset sizes.

* Section E.5.8 mentions that Influence Functions (i.e., LOO) took >24 hours on a small CNN model.  Having worked with influence functions before, including the source implementation you cite, this seems quite high.

* For TracIn, sampling every 15 epochs seems too sparse -- especially if you are training only 30 epochs and seems more akin to TracInCp than TracIn -- correct me if I misunderstood.  Also, reviewing Pruthi et al. again, using only the last linear layer is a speed-up due to limitations with Influence Functions.  Did you compare to TracIn using all layers as well?

    * Further to this end, it would be helpful if you had a table of the hyperparameters for the various methods (e.g., influence functions) to determine how the corresponding baseline hyperparameters were set.  That would allow me to better evaluate your setup.

* Since you are considering groups of points, perhaps second-order influence methods that directly consider group effects would have been a better baseline choice.  See for example: Basu et al. 2021 “On Second-Order Group Influence Functions for Black-Box Predictions” or Koh et al. 2019 “On the Accuracy of Influence Functions for Measuring Group Effects”.  If these baselines do not work, please provide more of an explanation why.

**Questions**

1) I assume $\mathcal{A}$ can be stochastic (e.g., in initial parameters, batch sequence, underlying implementation like CUDA).  Should Objective (1) then be the expected utility over randomization?  If Objective (1) is shown as deterministic for simplicity, that should be stated.

2) Did you observe randomization having any effect on $\hat{U}$’s performance (e.g., restarting analysis leading to meaningfully different utility predictions)?  As mentioned above, I did not see any ablation-like analysis of just this component of the method.

3) Do any alternatives exist to DeepSets (Zaheer et al 2017), and if so why did you select DeepSets over alternatives?

**General Points**

* In the abstract, you specify a “key insight”, but I was unsure of its intended meaning. It seemed like the most likely meaning was that the “bad data” does not contribute much (if anything) to a model’s corresponding performance metric(s).  This feels like an elementary insight so I think I may have missed your intended idea.

* You use the term “general robustness” in the introduction’s first paragraph.  When I read it, I was not sure exactly what you intended (it does becomes clear later).  General robustness is a clear strength of your method so being explicit on this definition earlier may be more impactful to a reader.

* A reference that seems related to your work is Feldman & Zhang from NeurIPS’20.

* I think your claims of general robustness would be significantly strengthened if you had more empirical evaluation on NLP data.  SPAM is used but the bag of words representation is somewhat simple and outdated.

* There is a good number of typographical errors in the main text.


**Summary Of The Paper:**

This paper proposes, *DataSifter*, an optimization-based, general-purpose framework for filtering "bad data" from a training set.  General-purpose broadly covers different data corruption types (e.g., adversarial perturbation, label noise, etc.), different model architectures, and performance metrics (e.g., test error).

**Summary Of The Review:**

Promising paper with clear ideas.  I have concerns in particular around the evaluation as detailed above.  Author response is important for my setting of the final score.

Edit: The authors provided thoughtful detailed reviews and addressed many of my concerns.  Their responsiveness and turnaround time on feedback was quite remarkable and duly noticed.

---

> ### Author Response · Authors · 2021-11-19
> **Response to Reviewer yuTL (Part 1)**
>
> ### Questions about Experiments (Part 1)
>
> **Q. [The setting of TracIn baseline can be made stronger]** *“For TracIn, sampling every 15 epochs seems too sparse -- especially if you are training only 30 epochs and seems more akin to TracInCp than TracIn -- correct me if I misunderstood. Also, reviewing Pruthi et al. again, using only the last linear layer is a speed-up due to limitations with Influence Functions. Did you compare to TracIn using all layers as well?”*
>
> **A.** We gratefully thank the reviewer for pointing out that the setting of TracIn in our experiment can be made stronger. **We have updated the implementation to match the exact TracIn instead of TracInCP described in the original paper. We have re-run ALL of the experiments where TracIn can apply and updated the figures in the modified version of the paper. Our algorithm still outperforms TracIn by a large margin for most of the settings**.
>
> Specifically, we trace the influence of data point z on the loss of z’ by summing over all iterations (not checkpoints) in which z is chosen in the batch. Besides, we use all layers in the neural networks. We also notice that the TracIn baseline can actually be separated into two variants for data selection tasks. **In the modified version of our paper, we replaced the original TracInCP baseline with two kinds of TracIn**: (1) TracIn-Clean which computes the influence on the loss change of clean validation data, and (2) TracIn-Self which traces the self-influence, i.e., the reduction of a training point on its own loss. TracIn-Self is used for bad data detection tasks, following the original paper.
>
> As we can see from the updated Figure 1 I and II, TracIn-Clean exhibits similar trends as influence function, which only starts finding the adversarially perturbed data points until filtering out a certain amount of clean data. We conjecture that this is because adversarially perturbed data points are designed to change the model prediction of particular target examples; each individual perturbed data point, however, does not have a significant impact on the loss of clean validation data. Besides, adversarially perturbed data points are similar and their influence on validation data is also similar in nature, thus they tend to be ranked together by influence-based methods. We can also see from other updated figures that under most of the scenarios, DataSifter outperforms both TracIn-Self and TracIn-Clean. This is mainly because influence-based methods are still linear heuristics which ignores the interaction between selected data points.
>
>
> **Q. [Poisoning rate in backdoor attacks are too large]**
>
> **A.** **We have re-run the experiment of Trojan attack on CIFAR-10 (main text I) where the poisoning rate is reduced to 5%**. This is a small poisoning rate since the attack success rate is around 90% and will drop quickly if we further reduce the poisoning rate (~75% for 3% and ~30% for 1% poisoning rate). **The updated experiment result is shown in Figure 2 I**. As we can see, in this case, DataSifter still significantly outperforms other baselines.
>
> **Q. [More baselines should be considered]** *”More baselines should be considered, e.g., Basu et al. 2020 “On Second-Order Group Influence Functions for Black-Box Predictions” or Koh et al. 2019 “On the Accuracy of Influence Functions for Measuring Group Effects”.”*
>
> **A.** We thank the reviewer for pointing out relevant works. **The influence function baseline in our paper is equivalent to [Koh et al. 2019]**, since their approach to estimate group influence is simply adding up all individual influences. Thus, to find the most influential groups with cardinality constraint $k$, one can just sort the training data by their influence on the validation set and pick the data points with top-$k$ influence. This is exactly what we did in the experiment. We have added this reference for influence function baseline.
>
> **[Basu et al. 2020] is not applicable** since the group selection method proposed in the paper is super expensive in addition to the expensive nature of the influence function. In their paper, they only tested on a toy synthetic dataset and toy model. For general neural networks (even for linear models), more efficient group selection algorithms would be required (**confirmed with the authors of [Basu et al. 2020]**). We have updated the baseline settings in the Appendix which briefly explains why this method is not applicable.
>
>
> **Q: [Correlation plot between predicted and actual subset utility]**
>
> **A.** Thanks for the useful suggestions! We have included examples of the correlation plot and added a section on the generalization of data utility learning in Appendix E5.1.
>
> **Q. [Table Summary of Hyperparameters]**
>
> **A.** Thanks for the useful suggestion! We have added Table 3 which summarizes the hyperparameter settings for baseline algorithms, and Table 4 summarizes the hyperparameter settings for bad data detection tasks.

---

> > ### Author Response · Authors · 2021-11-19
> > **Response to Reviewer yuTL (Part 2)**
> >
> > ### Questions about Experiments (Part 2)
> >
> > **Q: [The models used in the experiments are small]**
> >
> > **A:** In fact, the proposed technique is fully generalizable to large-size models via the existing proxy model techniques, e.g., see [LC94, CY+20, WCJ21]. The key idea is to use an efficient proxy model as a surrogate for the original, computationally intensive model to guide data selection. While these scaled-down models achieve significantly lower accuracy than larger models, the previous works find that they still provide useful information to rank and select points. Since the application of proxy models in data utility learning has already been studied empirically in the prior work, we decided not to include more experiments on it. We have updated the discussion about the proxy model technique to improve the scalability of DataSifter with more details at the end of Section 5.
> >
> > [LC94] Heterogeneous Uncertainty Sampling for Supervised Learning, ICML’94
> >
> > [CY+20] Selection via Proxy: Efficient Data Selection For Deep Learning, ICLR’20
> >
> > [WCJ21] One-round Active Learning, ArXiv
> >
> >
> > **Q: [Question about Mislabel Experiment]** *”In Mislabel Data Detection experiment (Figure 2 IV), why the change in the amount of mislabeled data does not affect validation accuracy too much?”*
> >
> > **A.** We conjecture that this is because for Enron SPAM dataset, the margin between the two classes is large so a small amount of mislabeled data points do not affect model accuracy too much (e.g., see the figure in https://ibb.co/D8BHMTR).
> >
> > We have updated the main text to better explain this phenomenon.
> >
> >
> > **Q. [For backdoor attack, how does the attack success rate being measured?]** *“Methods like TracIn and Influence Functions consider a specific target example. Are the attack success rate values w.r.t. to the specified target or any backdoored sample? How was the target selected?”*
> >
> > **A.** The attack success rate of backdoor attacks is measured by **a hold-out test backdoored dataset**, which is different from the set that TracIn and Influence function are calculated with respect to. We have updated the experiment settings to make this point clear.  For TracIn-Clean and Influence function, we sort all training examples by their influence on the clean validation set (examples with larger influence on the reduction of validation loss are considered as **clean data**). For TracIn-Self, we sort all training examples by their self-influence (examples with larger influence on the reduction of its own loss are considered as **bad data**). The attack access rate is calculated by a separate set that is unseen in the calculation of the TracIn-Clean and Influence function.
> >
> >
> > **Q. [Runtime of Influence Function]** *“Section E.5.8 mentions that Influence Functions (i.e., LOO) took >24 hours on a small CNN model.”*
> >
> > **A.** We think there is a misunderstanding that we would like to clarify. Influence function does terminate in 24 hours and it was included in the plot (denoted as 'INF'). 'LOO' refers to another baseline that computes the exact value of leave-one-out score. It does not terminate in 24 hours since the number of model retraining it requires is equal to the total number of the training set.
> >
> >
> > **Q. [Alternatives of DeepSets architecture?]** *“Do any alternatives exist to DeepSets (Zaheer et al 2017), and if so why did you select DeepSets over alternatives?”*
> >
> > **A.** As we mentioned in Section 7, there certainly exists other options of set learning architecture or algorithms, e.g., Set Transformer [Lee19] or Submodular Regularization [AA+20]. We use DeepSets because it is a canonical architecture for set function learning. Using more advanced set learning strategies is expected to produce stronger results. However, given that our scope is to devise a new data selection framework based on set learning and the framework instantiated with the canonical set function learning architecture can already evidence the advantage of the proposed framework, we decided to leave the exploration of other kinds of set learning architecture for data utility learning as future work.
> >
> > [Lee19] Set Transformer: A Framework for Attention-based Permutation-Invariant Neural Networks, ICML’19.
> >
> > [AA+20] Learning to make decisions via submodular regularization, ICLR’20
> >
> > **Q. [More empirical evaluation on NLP data?]**
> >
> > **A.** We agree that it is interesting to explore the application of the proposed algorithm to NLP domain. Given the diversity of NLP tasks (e.g., dialog system, translation, question answering) and the lack of prior work on the data utility learning for language modeling, a thorough investigation of DataSifter and its application to data selection in NLP is worth a paper in itself. Hence, we leave it for future exploration.

---

> > > ### Author Response · Authors · 2021-11-19
> > > **Response to Reviewer yuTL (Part 3)**
> > >
> > > ### Questions about Methods
> > >
> > > **Q: [Expanded discussion about approximate submodularity]** *“A more quantitative understanding/definition of what constitutes "approximate" would be appreciated. ... How does approximate submodularity affect DataSifter’s practical performance?”*
> > >
> > > **A.** Thanks for raising this question! We include a definition of approximate submodularity in Appendix E, which was originally proposed in [WYJ21]. It is stated as follows: a set function $f$ satisfies $\beta$-relaxed submodularity if for every $S \subseteq T \subseteq N$, and every $j \in N \setminus T$, we have $f(T \cup \{ j \}) - f(T) \le f(S \cup \{j\})-f(S) + \beta^{n-(|T|-|S|)}$ for some $0 < \beta < 1$. In [WYJ21]. The paper performed experiments on some common ML models such as logistic regression and several neural network models and showed that this definition can well describe the data utility functions for these modes. Besides, as we discussed in Appendix E, it turns out that functions that satisfy the above conditions belong to a larger function class called self-bounding functions, which has been proven to be efficiently learnable. This could serve as an insight into why data utility functions can be efficiently learned.
> > >
> > > Our experience is that the degree of data utility function’s approximate submodularity does not affect DataSifter’s practical performance too much empirically. For instance, when the learning algorithms are neural networks trained by stochastic gradient descent, there is huge randomness involved in the training process, and thus the utility samples may deviate a lot from submodularity. However, as we showed in the experiment, DataSifter works very well empirically even for deep learning tasks.
> > >
> > > [WYJ21] Learnability of Learning Performance and Its Application to Data Valuation, Arxiv
> > >
> > >
> > > **Q: [Can $\mathcal{A}$ (the learning algorithm) be stochastic? Will the randomness in $\mathcal{A}$ affect data utility learning?]**
> > >
> > > **A.** We appreciate the suggestion, and we have updated our definition of data utility functions $U$ in the main text, which takes the expectation over the randomness of the learning algorithm $\mathcal{A}$.
> > >
> > > **In the updated Appendix F5.1, we discuss the impact of randomness in learning algorithms on data utility learning and prediction**. When $\mathcal{A}$ is a randomized function, the utility sample $\{(S, u(\hat f, V))\}$ is a random variable, and the variance of $U_{\mathcal{A}, u}(S)$ will surely introduce more difficulty in data utility learning. However, as we show in Figure 5b (in Appendix), when the randomness of the learning algorithm is very large, data utility learning is still able to differentiate between subsets with very different expected utility.
> > >
> > >
> > > **Q. [Key insight behind DataSifter?]** *"Does the paper’s key insight “bad data does not contribute much (if anything) to a model’s corresponding performance metric(s)"?"*
> > >
> > > **A.** Yes, the reviewer’s understanding is correct. This is indeed a simple insight but the key to achieving “general robustness” in our paper. “Bad data” can have uncountably many possible formats (e.g., mislabel, noisy, adversarially perturbed), and their only common feature is that they contribute little to the model performance. Therefore, we exploit this common feature to tackle the problem of achieving general robustness for bad data.  We would like to highlight the simple insight is challenging to materialize due to the computational costs to estimate the learning performance for a combinatorial number of subsets. Our paper proposes a learning-based framework to efficiently estimate the performance for any given dataset and further perform dataset selection.
> > >
> > >
> > > **Q. [The definition of “General Robustness” needs to be clear in the first paragraph]**
> > >
> > > **A.** We have re-written the first paragraph of the paper.
> > >
> > > **Q. [The work by Fieldman and Zhang 2020 “What Neural Networks Memorize and Why: Discovering the Long Tail via Influence Estimation” might be relevant]**
> > >
> > > **A.** We thank the reviewer for pointing out the relevant work. We have included it as a related work of influence-based methods in the paper.

---

> > > > ### Author Response · Authors · 2021-11-27
> > > > **Message to Reviewer yuTL**
> > > >
> > > > Dear Reviewer yuTL,
> > > >
> > > > We'd like to express our gratitude once more for your constructive suggestions, which resulted in interesting revision updates. We've responded to each of your questions. Hopefully, you'll find that they adequately address your concerns. Additionally, we'd like to know if you have any additional questions or require clarification before the rebuttal phase concludes. We would be delighted to address them in the revision.
> > > >
> > > > Best wishes,
> > > >
> > > > Authors of Paper

---

> > > > > ### Comment · Reviewer_yuTL · 2021-11-30
> > > > > **Reviewer Response yuTL**
> > > > >
> > > > > I appreciate your detailed and lengthy response to my review.
> > > > >
> > > > > (1) (New) Figure 5 - This plot is useful and lends credibility to your overall argument.  I think it is worthwhile to include in the main paper in particular given the extended camera-ready length. (New) Section F.5.1's writing should be improved since there are a good deal of typos.  I understand it was written quickly for the updated submission.
> > > > >
> > > > > (2) Thank you for including the updated TracIn results.  With so many lines and similar colors in the plots, it is hard at time to distinguish one method from the other in particular for those of us with poorer eyesight.  This may not be remediable however.
> > > > >
> > > > > *  Note in Pruthi et al. 2020 misclassified training examples may be excluded since they can have strong negative effects on TracIn with cleaning.  It is not practicable to investigate that as a baseline now, but may be worth it for a stronger camera ready.
> > > > >
> > > > > (3) "The influence function baseline in our paper is equivalent to [Koh et al. 2019]" If this is the case, this was not clear in the manuscript.  I think citing the original Koh et al. 2017 if you are following Koh et al. 2019 confuses the matter.
> > > > >
> > > > > (4) "Q: [The models used in the experiments are small]" I understand your argument here, but it is less than satisfying.  I understand it may be challenging but to not include even a contrived experiment raises concerns for me.
> > > > >
> > > > > (5) "Q. [Key insight behind DataSifter?]" Thank you for the clarification.  As a point of feedback, I liked your explanation above and found it quite intuitive -- more so than the current text in the paper.
> > > > >
> > > > > I have not decided on a final score yet, and will do so after more discussion with the other reviewers.

---

> > > > > > ### Author Response · Authors · 2021-11-30
> > > > > > **Followup Response to Reviewer yuTL**
> > > > > >
> > > > > > We want to express our gratitude once more for the useful comments.
> > > > > >
> > > > > > **Q. [Large-Model Experiment with Proxy Model Technique?]**
> > > > > >
> > > > > > **A:** We appreciate the feedback for including large-model experiments. However, we believe that our existing results can already demonstrate the potential of DataSifter’s utility for large models with the previously mentioned proxy model technique. Note that all of our results regarding bad data detection are evaluated in terms of **bad data detection rate** (which is a **model-agnostic** measure of robustness) and our method outperforms the other baselines in terms of detection rate. It is reasonable to expect that using a high-detection-rate method as a preprocessing step should eventually lead to a model with higher performance, no matter how large the model is.
> > > > > >
> > > > > > To corroborate the point above, we performed another experiment overnight on noisy data detection for a large model (same setting as in Appendix F5.9). The result shows that because of the high detection rate of our method, large models can also achieve good performance when trained on the data sanitized by our method. The model we used here is ResNet18, which is significantly larger than the CNN model we used in the original experiment. We didn’t use even larger models such as ResNet50 due to limited time. In this case, the small CNN model we used in the previous experiment serves as a proxy model for the ResNet18 model, and we select data points by using the utility model trained on the utility samples collected by repeated training the small CNN models. All other baselines are directly run on the ResNet model if its computation depends on the underlying model (influence function, tracin-clean and tracin-self). We only run the influence function for one random seed since it almost takes 10 hours to finish. The result of noisy data detection rate is in https://ibb.co/xS8f0d4, and the result of model accuracy vs thrown data is in https://ibb.co/cXDjHhz. As we can see, DataSifter once again significantly outperforms all other baselines on the large model.
> > > > > >
> > > > > > Overall, we hope the above quick experiment can showcase that DataSifter can be easily extended to large models with the previously mentioned proxy model technique. Secondly, because of the high detection rate achieved by using simple models (as evidenced by our existing experiments), the advantage of our method would remain on large models.
> > > > > >
> > > > > >
> > > > > >
> > > > > >
> > > > > >
> > > > > >
> > > > > > **Q. [Stronger baseline for TracIn by excluding misclassified training examples?]** *“Note in Pruthi et al. 2020 misclassified training examples may be excluded since they can have strong negative effects on TracIn with cleaning.”*
> > > > > >
> > > > > > **A:** If our understanding is correct, you are suggesting that we could first train once on the full training set, remove those that are misclassified, and then perform TracIn on the remaining data points for a better performance. We appreciate this pointer and would be happy to include this “enhanced” version of TracIn into the final version. However, we do think that this “enhanced” version would not necessarily enhance the performance of data selection, because of the subtleties of whether the misclassified data shall be categorized into good (i.e., retaining the data in the training set) or bad data (i.e., excluding the data from the training set). Note that the misclassified examples are excluded without having the utility or influence being evaluated; hence, there is not enough information on whether these misclassified data are more likely to be good or bad. If we count misclassified data as bad data, then these samples would take the removal budget for real bad data detected based on utility or influence (i.e., mislabeled, poisoned data, etc), hence hurting the robustness; however, if we count misclassified data as good,  then the model accuracy may degrade. It is foreseeable that this “enhanced” method does not lead to a better tradeoff between robustness and accuracy. Moreover, we would like to emphasize that for many of the experiment settings, the misclassification rate is very small (e.g., for MNIST and SPAM dataset); but on the other hand, the current version of TracIn underperforms the other baselines by a considerable margin. Hence, it is very likely that the “enhanced” version of TracIn by excluding misclassified data points still would not outperform the other baselines.
> > > > > >
> > > > > > Please correct us if we have any misunderstanding about your comment, and we highly appreciate any further clarification on this point.

---

> > > > > > > ### Comment · Reviewer_yuTL · 2021-12-03
> > > > > > > **Updated Score**
> > > > > > >
> > > > > > > Dear Authors, I appreciate your overall responsiveness throughout this dialogue.  Your quick turnaround on the ResNet18 experiments assuages some of my concerns.  I am admittedly not an expert on ICLR's exact rules on posting new data at this stage.  Nonetheless, I reviewed the plots you provided and updated my score accordingly.
> > > > > > >
> > > > > > > On the latter point, TracIn's performance may look artificially poor when analysis considers mislabeled training instances since those tend to look highly influential.  It's a side issue at this point.  I wanted to make sure I acknowledged that I read and considered your latest response.

---

> > > > > > > > ### Author Response · Authors · 2021-12-03
> > > > > > > > **Sincerely Thankful for Raising the Score, and Follow-up Response About TracIn**
> > > > > > > >
> > > > > > > > We sincerely thank the reviewer for raising the score!
> > > > > > > >
> > > > > > > > For your comment about the potentially stronger baseline for TracIn by excluding mislabeled training examples, we believe you refer to “misclassified” training instances instead of “mislabeled” in the last post. We understood your concern about the impact of misclassified data on the performance of TracIn. We apologize for not explaining this point as clearly as we intended. So, we decided to run an experiment overnight to use empirical results to make further clarification and the result shows that filtering out misclassified training instances does NOT provide large improvement for TracIn. Due to time constraints, we conduct the experiment of backdoor detection on MNIST. Specifically, we tested 4 additional variants of TracIn, depending on whether we treat the filtered misclassified data points as good or bad data:
> > > > > > > > - ‘TracIn-Self+GOOD’: we filter out misclassified data points for calculating influence but retain them in the final training set (i.e., misclassified data are counted as **good** data), and the influence score is computed by TracIn-Self.
> > > > > > > > - ‘TracIn-Self+BAD': we filter out misclassified data points for calculating influence and also exclude them from the final training set (i.e., misclassified data are counted as **bad** data), and the influence score is computed by TracIn-Clean.
> > > > > > > > - ‘TracIn-Clean+GOOD’: we filter out misclassified data points for calculating influence but retain them in the final training set (i.e., misclassified data are counted as **good** data), and the influence score is computed by TracIn-Clean.
> > > > > > > > - ‘TracIn-Clean+BAD': we filter out misclassified data points for calculating influence and also exclude them from the final training set (i.e., misclassified data are counted as **bad** data), and the influence score is computed by TracIn-Clean.
> > > > > > > >
> > > > > > > > The model training accuracy is 94.3%, which means that there are 57 misclassified training instances, and we filter them out from the calculation of influence (i.e., TracIn-Self or TracIn-Clean). In this experiment, all 57 misclassified training instances are clean data since the model can easily perform well on backdoored data, so ‘+GOOD’ variants should perform better in this case. Since the misclassified examples are excluded without having the utility or influence being evaluated, in the plot of Detection Rate vs # Inspected Data, we randomly inspect data points from misclassified examples at the beginning (if counted as bad data) or at the end (if counted as good data).
> > > > > > > >
> > > > > > > > For visibility, we only show curves for DataSifter and all of the TracIn techniques. The result of backdoor data detection rate is in https://ibb.co/z6jZ23Z, and the result of attack accuracy vs thrown data is in https://ibb.co/vQFkRH4. As we can see, the variants do not significantly improve the performance of TracIn, and the slight improvement is mainly due to ‘+GOOD’ variants correctly identifying all misclassified training instances as good data. **However, this might not be the case for other kinds of bad data types.** For example, for noisy data experiment, we find that most of the misclassified data are bad data. Thus, it’s not clear whether to use ‘+GOOD’ variant or ‘+BAD’ variant without knowing the specific data issue type. This fact that the best version of TracIn implementation depends on the underlying bad data type deviates from our original goal of designing data selection algorithm that achieve *“general robustness.”*
> > > > > > > >
> > > > > > > > Overall, this experiment evidences the point that we made in the previous post: removing misclassified data from TracIn will not make it outperform the other baselines because (1) TracIn mostly underperforms the other baselines by a large margin and (2) The misclassified data points are either removed or retained without their utility being evaluated and hence will **waste** the budget of removing real bad data or retaining real good data.

---

### Official Review · Reviewer_GEXW · 2021-11-05

**Correctness:** 2
**Technical Novelty And Significance:** 2
**Empirical Novelty And Significance:** 2
**Recommendation:** 5
**Confidence:** 3

**Main Review:**

Overall the paper is clearly written and presents a novel framework to identify bad training data. Empirical results and comparison with baselines look interesting.

However, I have the following concerns with the work:

-  Proposed method (as described in Section 5) is based on a set learning phase where the task is the following: Given a training set, the aim is to learn how an underlying model when trained on that training set will generalize to unseen data from the same distribution. Their method DATASIFTER crucially depends on this step. However, this set learning problem is in general underspecified and hence hard to tackle. At a high level, to perform well on this task, the set functional learning architecture would not only need to mimic the underlying model trained on some training set (without actually using that model) but would also need to understand how the underlying model generalizes. The authors present no empirical evidence their algorithm solves this task in a reasonable manner.  Some empirical results showing correlations would be interesting to see.

- Models considered in the work are small in size as compared to the state-of-the-art models. Is there any specific reason for this?

**Summary Of The Paper:**

This paper focuses on the problem of identifying bad training data when the underlying cause is unknown in advance. Authors develop an algorithmic framework, DATASIFTER, for general robustness to bad training data. Empirical evaluation show efficacy of DATASIFTER in a wide range of tasks, including backdoor, poison, noisy/mislabel data detection, data summarization, and data debiasing

**Summary Of The Review:**

The proposed framework of DATASIFTER is interesting and novel with promising results. However, the method primarily depends on a set selection phase for which no empirical/theoretical evidence is provided.

---

> ### Author Response · Authors · 2021-11-19
> **Response to Reviewer GEXW**
>
> **Q1: [Empirical evidence for learnability of data utility]** *”Some empirical results showing correlations between predicted and actual data utility would be interesting to see.”*
>
> **A:** We have added the empirical evidence for the generalization of data utility models into Appendix F5.1. The result shows that data utility learning with set functions exhibits good generalizability.
>
> Besides that, we have added a discussion about the learnability of data utility functions in Section E. We agree with the reviewer that arbitrary set functions are not efficiently learnable. However, as we mentioned in the paper, recent works show that data utility functions for many common ML algorithms exhibit "approximate" submodularity [WCJ21, WYJ21]. It is known in the literature that functions that satisfy "approximate" submodularity (precise definition in the Appendix) can be efficiently learned under mild conditions. This could serve as an insight into why we can learn many data utility functions efficiently.
>
> [WCJ21] One-round Active Learning, ArXiv
>
> [WYJ21] Learnability of Learning Performance and Its Application to Data Valuation, ArXiv
>
> **Q2: [Models used in the experiments are small compared to state-of-the-art]**
>
> **A:** In fact, the proposed technique is fully generalizable to large-size models via the existing proxy model techniques, e.g., see [LC94, CY+20, WCJ21]. The key idea is to use an efficient proxy model as a surrogate for the original, computationally intensive model to guide data selection. While these scaled-down models achieve significantly lower accuracy than larger models, the previous works find that they still provide useful information to rank and select points. Since the application of proxy models in data utility learning has already been studied empirically in the prior work, we decided not to include more experiments on it. We have updated the discussion about the proxy model technique to improve the scalability of DataSifter with more details at the end of Section 5.
>
> [LC94] Heterogeneous Uncertainty Sampling for Supervised Learning, ICML’94
>
> [CY+20] Selection via Proxy: Efficient Data Selection For Deep Learning, ICLR’20
>
> [WCJ21] One-round Active Learning, ArXiv

---

> > ### Author Response · Authors · 2021-11-27
> > **Message to Reviewer GEXW**
> >
> > Dear Reviewer GEXW,
> >
> > We'd like to express our gratitude once more for your constructive suggestions, which resulted in interesting revision updates. We've responded to each of your questions. Hopefully, you'll find that they adequately address your concerns. Additionally, we'd like to know if you have any additional questions or require clarification before the rebuttal phase concludes. We would be delighted to address them in the revision.
> >
> > Best wishes,
> >
> > Authors of Paper

---

> > > ### Author Response · Authors · 2021-12-03
> > > **Follow-up Response to Reviewer GEXW**
> > >
> > > Dear Reviewer,
> > >
> > > We want to send a gentle reminder regarding your concern about using small models in the experiment. Since Reviewer yuTL shares the same concerns as you, we have performed additional experiments to show that our method is fully generalizable to large models through the proxy model technique.
> > >
> > > Here is our follow-up response to Reviewer yuTL, which has already addressed his/her concern. We copied the response here for your reference, and hopefully it can also adequately address your concern.
> > >
> > > **Q. [Large-Model Experiment with Proxy Model Technique?]**
> > >
> > > **A:** We appreciate the feedback for including large-model experiments. However, we believe that our existing results can already demonstrate the potential of DataSifter’s utility for large models with the previously mentioned proxy model technique. Note that all of our results regarding bad data detection are evaluated in terms of **bad data detection rate** (which is a **model-agnostic** measure of robustness) and our method outperforms the other baselines in terms of detection rate. It is reasonable to expect that using a high-detection-rate method as a preprocessing step should eventually lead to a model with higher performance, no matter how large the model is.
> > >
> > > To corroborate the point above, we performed another experiment overnight on noisy data detection for a large model (same setting as in Appendix F5.9). The result shows that because of the high detection rate of our method, large models can also achieve good performance when trained on the data sanitized by our method. The model we used here is ResNet18, which is significantly larger than the CNN model we used in the original experiment. We didn’t use even larger models such as ResNet50 due to limited time. In this case, the small CNN model we used in the previous experiment serves as a proxy model for the ResNet18 model, and we select data points by using the utility model trained on the utility samples collected by repeated training the small CNN models. All other baselines are directly run on the ResNet model if its computation depends on the underlying model (influence function, tracin-clean and tracin-self). We only run the influence function for one random seed since it almost takes 10 hours to finish. The result of noisy data detection rate is in https://ibb.co/xS8f0d4, and the result of model accuracy vs thrown data is in https://ibb.co/cXDjHhz. As we can see, DataSifter once again significantly outperforms all other baselines on the large model.
> > >
> > > Overall, we hope the above quick experiment can showcase that DataSifter can be easily extended to large models with the previously mentioned proxy model technique. Secondly, because of the high detection rate achieved by using simple models (as evidenced by our existing experiments), the advantage of our method would remain on large models.

---

### Author Response · Authors · 2021-11-19
**Summary of changes in rebuttal revision**

We thank all of the reviewers for the detailed and valuable comments. We are glad that our work is recognized as novel and important (GEXW, yuTL, 4LgM), well-written (GEXW, yuTL, 4LgM), with solid experimental evaluation (yuTL, 4LgM, s9Xw).

We studied the reviews and discussions carefully and modified our paper accordingly. All modifications are highlighted. Here’s a summary of our major revision to the paper.

**Introduction (Section 1)**: we re-written the first paragraph of the paper.

**Formulation (Section 3):** we changed the definition of data utility function to an expectation over the randomness of the learning algorithm.

**Method (Section 5)**: in the first paragraph of the section, we explicitly discussed the worst-case analysis of DataSifter under the full recovery assumption and unbounded computation budget.

**Method (Section 5)**: at the end, we updated the discussion about the proxy model technique to improve the scalability of DataSifter with more details.

**Experiment (Section 6)**: we changed TracIn baseline to TracIn-Self and TracIn-Clean.

**Experiment (Section 6)**: we reduced the poisoning rate of backdoor attacks I and re-run the experiment.

**New Appendix E**: we added a discussion about the characterization and learnability of data utility functions.

**New Appendix F1**: we added the description of TracIn-Clean and TracIn-Self baselines; table summary of baseline settings.

**New Appendix F3**: we added a table summary of hyperparameters in Bad Data Detection experiments.

**New Appendix F5.1**: we added empirical results and discussion of generalization of data utility learning.

---

### Decision · Program_Chairs · 2022-01-20

**Decision:**

Reject

**Comment:**

The paper considers the question of identifying bad data so that models can be trained on the subset of data that is good. This question is formulated as a utility optimization problem. The paper shows that some popular heuristics are quite bad in the framework they propose. They also propose a new algorithmic framework called DataSifter. There is empirical evaluation provided for this. Questions have been raised in the reviews about the size of the models that have been used in the empirical evaluation. The authors have responded to this by suggesting the use of proxy model techniques. There are also questions about learnability of data utility for which some responses are provided in the rebuttal.